

# The sensitivity of primary productivity in Disko Bay, a coastal Arctic ecosystem to changes in freshwater discharge and sea ice cover

Eva Friis Møller[1], Asbjørn Christensen[2], Janus Larsen[1], Kenneth D. Mankoff[3,4], Mads Hvid Ribergaard[5], Mikael Sejr[1], Philip Wallhead[6], Marie Maar[1]

[1]Department of Ecoscience, Aarhus University, 4000 Roskilde, Denmark

[2]DTU Aqua, Technical University of Denmark, DK-2880 Kgs. Lyngby, Denmark

[3]Department of Glaciology and Climate, Geological Survey of Denmark and Greenland, 1350 Copenhagen, Denmark

[4]National Snow and Ice Data Center (NSIDC), Cooperative Institute for Research in Environmental Sciences (CIRES), University of Colorado Boulder, Boulder, CO, 80390, USA

[5]Danish Meteorological Institute, 2100 Copenhagen, Denmark

[6]Section for Oceanography, Norwegian Institute for Water Research (NIVA Vest), Bergen, Norway

*Correspondence to*: Eva Friis Møller (efm@ecos.au.dk)





**Abstract.** The Greenland Ice Sheet is melting, and the rate of ice loss has increased 6-fold since
the 1980s. At the same time, the Arctic sea ice extent is decreasing. Melt water runoff and sea ice
reduction both influence light and nutrient availability in the coastal ocean with implications for
the timing, distribution and magnitude of phytoplankton production. However, the integrated
effect of both glacial and sea ice melt is highly variable in time and space, making it challenging
to quantify. In this study, we evaluate the relative importance of these processes for the primary
productivity of Disko Bay, West Greenland, one of the most important areas for biodiversity and
fisheries around Greenland. We use a high-resolution 3D coupled hydrodynamic-biogeochemical
model for 2004 to 2018 validated against *in situ* observations and remote sensing products. The
model estimated net primary production (NPP) varied between 90-147 gC $m^{-2}$ $year^{-1}$ during
2004-2018, a period with variable freshwater discharges and sea ice cover. NPP correlated
negatively with sea ice cover, and positively with freshwater discharge. Fresh water discharge
had a strong local effect within ~25 km of the source sustaining productive hot spot during
summer. When considering the annual NPP at bay scale, sea ice cover was the most important
controlling factor. In scenarios with no sea ice in spring, the model predicted ~30% increase in
annual production compared to a situation with high sea ice cover. Our study indicates that
decreasing ice cover and more freshwater discharge can work synergistically and will likely
increase primary productivity of the coastal ocean around Greenland.



## 1    Introduction

The warming of the Arctic (Cohen et al., 2020) has a strong impact on the regional sea ice. Over the past few decades, the sea ice melt season has lengthened (Stroeve et al., 2014), summer extent has declined, and the ice is getting thinner (Meier et al., 2014). This has an immediate effect on the primary producers of the ocean. The photosynthetic production is constrained by the annual radiative cycle, and the sea ice reduces the availability of light and thereby the development of the sea ice algae and the pelagic phytoplankton communities (Ardyna et al., 2020). An extended open water period will affect the phenology of primary producers and potentially lead to an earlier spring bloom (Ji et al., 2013; Leu et al., 2015), and may also increase the potential for autumn blooms (Ardyna et al., 2014).

In the Arctic coastal ocean, there are additional impacts of a warming climate. As the freshwater discharge increases due the melt of snow and ice on land and higher precipitation (Kjeldsen et al., 2015; Mankoff et al., 2020a, 2021), the land-ocean coupling along the extensive Arctic coastline is intensified (Hernes et al., 2021). The summer inflow of melt water has complex biogeochemical impacts on the coastal ecosystem and combines with changes in sea ice cover to affect the magnitude and phenology of marine primary production. In areas dominated by glaciated catchments such as Greenland, the increase in melt water discharge has been substantial and the rate of ice mass loss has increased sixfold since the 1980s (Mankoff et al., 2020b; Mouginot et al., 2019).

The changes in sea ice cover and freshwater discharge will affect the marine primary production through the complex interactions of changes in stratification, light and nutrient availability (Arrigo and van Dijken, 2015; Hopwood et al., 2020). The individual processes are relatively well described, but the interactions between them and the temporal and spatial importance under different Arctic physical regimes are less well understood.  A lower extent of sea ice cover may also increase the wind induced mixing of the water column and deepen or weaken the stratification. Thereby, the potential for the phytoplankton to stay and grow in the illuminated surface layer is reduced. At the same time, a higher mixing rate will increase the supply of new nutrients from deeper layers to support production when light is not limiting (Tremblay and Gagnon, 2009). Another mechanism affecting stratification is the freshening of the surface layer due to ice melt from both sea ice and the ice sheet  (von Appen et al., 2021; Holding et al., 2019).





However, if a glacier terminates in a deep fjord, the ice sheet melt is injected at depth causing
more coastal upwelling of nutrients before acting to increase surface layer stratification
(Hopwood et al., 2018; Meire et al., 2017)
The relative importance on productivity of sea ice versus glacier freshwater discharge depends
on the scale considered (Hopwood et al., 2019). Freshwater discharge from the ice sheet is more
important in the vicinity of the glacier (Hopwood et al., 2019; Meire et al., 2017), whereas the
sea ice dynamics are considered to be an important driver in the open ocean (Arrigo and van
Dijken, 2015; Massicotte et al., 2019; Meier et al., 2014). Most studies consider one or the other
separately (e.g. Hopwood et al., 2018; Vernet et al., 2021). However, in the coastal Arctic areas
at the mesoscale, i.e. 10-100 km, it can be expected that both sea ice and glacier freshwater
discharge and the interaction between them will influence the ecosystem and the pelagic primary
production (Hopwood et al., 2019). To resolve their relative impacts, we need to constrain their
impact on both seasonal and spatial scales, which is a challenging task. A useful tool to achieve
such an integrated perspective is a high-resolution 3D coupled hydrodynamic-biogeochemical
model.
Disko Bay is located on the west coast of Greenland (Fig. 1) near the southern border of the
maximum annual Arctic sea ice extent, and is influenced by both sub-Arctic waters from
southwestern Greenland and Arctic waters within the Baffin Bay (Gladish et al., 2015; Rysgaard
et al., 2020). The bay has a pronounced seasonality in sea ice cover (Møller and Nielsen, 2020).
Over the last 40 years, there has been pronounced decrease in sea ice cover, and also the year-to-
year variations have increased in the last decade (Fig 2, Hansen et al., 2006, the Greenland
Ecosystem monitoring program, http://data.g-e-m.dk). For the primary producers particularly the
decrease in sea ice cover during the time of the spring bloom in April is important (Møller and
Nielsen, 2020). In addition to the seasonal sea ice cover changes, the bay also experiences large
seasonal changes in freshwater input from the Greenland ice sheet, particularly during the
summer months (Fig. 2, 3).  The large marine terminating glacier Sermeq Kujalleq (Jakobshavn
Isbræ) is found in the inner part of the bay. It is estimated that about 10% of the icebergs from
the Greenland ice sheet originate from this glacier (Mankoff et al., 2020a). Since the 1980s,
freshwater discharge from the Greenland Ice sheet to Disko Bay has almost doubled (Fig. 2,
(Mankoff et al., 2020b, 2020a). How these significant changes in sea ice dynamics and run-off



will impact the ecosystem in Disko Bay, one of the most important areas for biodiversity and
fisheries around Greenland (Christensen et al. 2012), is still not well understood.
In this study, we investigate the combined effect of changes in sea ice cover and the Greenland
ice sheet freshwater discharge on the phenology/seasonal timing and annual magnitude and
spatial distribution of the phytoplankton production in Disko Bay. We do so using a high-
resolution 3D coupled hydrodynamic-biogeochemical model validated against in situ
measurement of salinity, temperature, nutrients, phytoplankton, and zooplankton biomass. The
validated model allows us to estimate the impact of sea ice cover and freshwater discharge on
productivity with a higher temporal and spatial resolution than would be possible from
measurements alone.

## 2   Methods


### 2.1   Hydrodynamic model


The model was set up using the FlexSem model system (Larsen et al. 2020). FlexSem is an open
source modular framework for 3D unstructured marine modelling
(https://marweb.bios.au.dk/flexsem). The system contains modules for hydrostatic and non-
hydrostatic hydrodynamics, 3D pelagic and 3D benthic models, sediment transport and agent-
based models. The source code can be found at the FlexSem webpage.
Bathymetry were obtained from the150x150 m resolved IceBridge BedMachine Greenland,
Version 3 (https://nsidc.org/data/IDBMG4 (Morlighem et al., 2017)) and interpolated to the
FlexSem computational mesh using linear interpolation. The 96,300 km$^2$ large computational
mesh for the Disko Bay area was constructed using the mesh generator JigSaw
(https://github.com/dengwirda/jigsaw) (Fig. 1). It consists of 6349 elements and 34 depth z-
layers with a total of 105678 computational cells. The horizontal resolution varies from 1.8 km
in the Disko Bay proper, 4.7 km in Strait of Vaigat and 16 km towards the semi-circular Baffin
Bay open boundary. In the deepest layers, the vertical resolution is 50 m, decreasing towards the
surface, where the top 5 layers are 3.5, 1.5, 2.0, 2.0 and 2.0 meters thick, respectively. The
surface layer thickness is flexible allowing changes in water level e.g., due to tidal elevations.
The model time step is 300 seconds and has been run for the period from 2004 to 2018.





## 2.2   Biogeochemical model


The biogeochemical model in the FlexSem framework was based on a modification of the
ERGOM model that originally was applied to the Baltic Sea and the North Sea (Maar et al.,
2011, 2016; Neumann, 2000) (Appendix A). In the Disko Bay version, 11 state variables
describe concentrations of four dissolved nutrients ($NO_3$, $NH_4$, $PO_4$, $SiO_2$), two functional groups
of phytoplankton (diatoms, flagellates), micro- and mesozooplankton, detritus (NP), detritus-
silicon, and oxygen. Cyanobacteria present in the Baltic Sea version of the model are removed in
the current set-up, because cyanobacteria are of little importance in high-saline Arctic waters
(Lovejoy et al., 2007). Further, pelagic detrital silicon was added to better describe the cycling
and settling of Si in deep waters. The model currency is N using Redfield ratios to convert to P
and Si. Chlorophyll *a* (Chl *a*) was estimated as the sum of the two phytoplankton groups
multiplied by a factor of 1.7 mg-Chl/mmol-N (Thomas et al., 1992). The calanoid copepod *C.*
*finmarchicus* generally dominates the mesozooplankton biomass (Møller and Nielsen, 2020) and
the physiological processes were parameterized according to previous studies (Møller et al.,
2012, 2016). The model considers the processes of nutrient uptake, growth, grazing, egestion,
respiration, recycling, mortality, particle sinking and seasonal mesozooplankton migration in the
water column and overwintering in bottom waters. NPP was estimated as daily means of
phytoplankton growth after subtracting respiration and integrated over 30 m depth corresponding
to the productive layer. The timing of the seasonal C. *finmarchicus* migration was calibrated
against in situ measurements of their vertical distribution over time (Møller and Nielsen, 2019).
Light attenuation (kd) is a function of background attenuation (water turbidity, kdb) and
concentrations of detritus and Chl *a* (Maar et al., 2011). Turbidity is strongly correlated with
salinity and the background attenuation was described as a function of salinity: kdb=0.80-salinity
x 0.0288 for salinity < 25 and a constant of 0.08 $m^{-1}$ for salinity >25 according to monitoring
data in the Disko Bay 69° 14' N, 53° 23' W (data.g-e-m.dk) and measurements across a salinity
gradient in another Greenland fjord, the Young Sound (Murray et al., 2015). Light optimum was
changed for both phytoplankton groups during calibration to fit with the timing of the spring
bloom (Appendix A). Background mortality of microzooplankton was increased to account for
other grazing pressure than from *C. finmarchicus*.



## 2.3 Freshwater and nutrient discharge

We used the MAR and RACMO regional climate model (RCM) runoff field to compute freshwater discharge. Ice runoff is defined as ice melt + condensation – evaporation + liquid precipitation – refreezing. Land runoff is computed similarly, but there is no ice melt term (although there is snow melt). Daily simulations of runoff were routed at stream scale to coastal outlets, where it is then called 'discharge'. Precipitation onto the ocean surface is not included in the calculations (Mankoff et al., 2020a) Within Disko Bay, 235 streams discharge liquid water, of which 97.5 % of the water comes from just 30 streams.

Fourteen points were selected within the model domain to represent the freshwater inflow. The locations were manually selected to best represent the location of the largest rivers and the spatial distribution of freshwater inflow in the model domain. The inflow from the 30 largest rivers were manually aggregated into the 14 point sources by evaluating the geographical location in relation to the coastal layout. This land run-off was inserted into the nearest model cell in the surface layer. Although subglacial discharge enters at depth, it rises up the ice front within a few 10s to 100s of meters of the ice front and within the grid cell at the ice boundary will reach its neutral isopycnal here assumed to be the surface layer (Mankoff et al., 2016). Thus, ice runoff were inserted in the surface layer. Solid ice discharge was computed from ice velocity, ice thickness, and ice density at marine terminating glaciers (Mankoff et al., 2020b). Within our modelling area in Disko Bay four glaciers discharge icebergs into fjords, of which the majority comes from Sermeq Kujalleq (Jakobshavn Isbræ). Solid ice was inserted where glaciers terminate directly into fjords (Fig. 1). At these four localities with marine terminating, the freshwater contribution as solid ice was assumed to be equally distributed in the top 100 m assuming that the majority of the solid ice are small pieces that melts quickly as evidenced by the lack of brash ice generally seen in Disko Bay. Thus, we do not consider the large icebergs calved by Sermeq Kujalleq and their input of freshwater along the route in the bay. Land discharge of nitrate, phosphate, and silicate at the 14 point sources was assumed to be constant in time with concentrations of 1.25, 0.20 and 10.88 mmol m$^{-3}$, respectively (Hopwood et al., 2020).

## 2.4 Hydrodynamic open boundary and initial data

At the semi-circular open boundary towards the Baffin Bay, the model was forced with ocean velocities, water level, salinity, and temperature obtained from a coupled ocean- and sea-ice





model (Madsen et al., 2016) provided by the Danish Meteorological Institute (DMI). The DMI
model system consists of the HYbrid Coordinate Ocean Model (HYCOM, e.g., Chassignet et al.,
2007) and the Community Ice CodE (CICE, (Hunke, 2001; Hunke and Dukowicz, 1997) coupled
with the Earth System modeling Framework (ESMF) coupler (Collins et al., 2005). The
HYCOM-CICE set-up at DMI covers the Arctic Ocean and the Atlantic Ocean, north of about
20°S, with a horizontal resolution of about 10 km. Further details on the HYCOM-CICE model
system can be found in Appendix B.
The 2D (water level) and 3D parameters were interpolated to match the open boundary in the
FlexSem Model setup using linear interpolation. Correspondingly, initial fields of temperature,
salinity and water level were interpolated from the HYCOM-CICE model output.

### 2.5 Observed sea ice cover

The long term sea ice cover within Disko Bay was extracted from the sea-ice concentration data
provided by the EUMETSAT Ocean and Sea Ice Satellite Application Facility (OSISAF,
www.osi-saf.org, Lavergne et al., 2019) on a daily basis (AICE). The Disko Bay area is here
defined as longitude and latitude range between 54.0°W and 51.5°W and 68.7°N to 69.5°N
respectively. As the OSISAF product is seasonally quite noisy for low sea ice concentrations, we
made a cutoff at 40 percent before we take the mean for the entire area. The exact cut-off value
does not matter much on the resulting time series, as the freeze-up and melt-down period is quite
fast for the area. Furthermore, we obtained sea ice observations from the Greenland Ecosystem
Monitoring (GEM) program (http://data.g-e-m.dk) in which ice coverage is registered daily by
visual inspection from the laboratory building at Copenhagen University's Arctic station in
Qeqertarsuaq.

### 2.6 Surface forcing data

At the surface, the model was forced by sea ice concentration, wind drag and heat fluxes. The ice
cover percentage modifies the wind drag, heat balance and light penetration in the model. The
surface heat budget model estimating the heat flux (long- and short-wave radiation) was forced
by wind, 2 meter atmospheric temperature, cloud cover, specific humidity and ice cover.
Photosynthetically active radiation (PAR) was estimated from the short-wave radiation assuming
43% to be available for photosynthesis (Zhang et al., 2010). The atmospheric forcing was
provided by DMI from the HIRLAM (Yang et al., 2005) and HARMONIE (Yang et al., 2017;



2018) meteorological models using the configuration with the best resolution available for our
simulation period. The resolution was 15 km until May 2005, then increased to about 5 km until
March 2017, and since then to 2.5 km. Ice cover was obtained from the HYCOM-CICE model
output.

### 2.7    Biogeochemical open boundary and initial data

Initial data and open boundary conditions for ecological variables were obtained from the pan-
Arctic 'A20' model at NIVA Norway. This was based on a 20 km-resolution ROMS ocean-sea-
ice model (Shchepetkin and McWilliams, 2005, Roed et al., 2014) coupled to the ERSEM
biogeochemical model (Butenschön et al., 2016), run in hindcast mode and bias-corrected
towards a compilation of in situ observations (Palmer et al., 2019). This model provided bias-
corrected output for (nitrate, phosphate, silicate, dissolved oxygen) plus raw hindcast output for
ammonium, detritus (small, medium and large fractions), 6 groups of phytoplankton and 3
zooplankton groups. The picophytoplankton, Synechococcus, nano-, micro-phytoplankton and
prymnesiophyte biomasses from ERSEM were summed to provide data for the autotrophic
flagellate group in ERGOM, while the diatom functional group was the same in both models.
The detritus pool in ERGOM was the sum of the three detritus size fractions in ERSEM.  The
A20 data were provided as weekly means on a 20 km grid and linearly interpolated to the
FlexSem grid. ERSEM provided data through 2014, then 2014 was repeated for the following
years.

### 2.8    Validation

For model calibration and validation of the seasonality, we used reported research observations
of temperature, salinity, nutrients (nitrate, silicate, phosphate), Chl *a* concentrations and
mesozooplankton biomass collected during short-term field campaigns at the Disko Bay station
69° 14' N, 53° 23' W from 2004 to 2012 (e.g.(Møller and Nielsen, 2019)). Furthermore, we used
observations of the same variables from the same station provided by the Greenland Ecological
Monitoring (GEM) program running since 2016 in the Disko Bay (data.g-e-m.dk). However, the
data coverage is highly sporadic between years and months, and we therefore created a monthly
climatology (2004-2018) for the best-sampled depth layer 0-20 m. This climatology was
compared with monthly means extracted from the model at the same location and depth range
where 2004 was used for model calibration and means from 2005 to 2018 for model validation.



Mesozooplankton biomass in the model was assumed to mainly represent the copepods *Calanus*
spp. and for the conversion from N to carbon (C) biomass, we used 12 g-C mol⁻¹ and C:N= 6.0
mol-C mol-N⁻¹ (Swalethorp et al., 2011).
Additionally, the model was validated spatially using remote sensing (RS) data of sea surface
temperature (SST) and Chl *a* concentrations for spring (April to June) and summer (July to
September) for 2010 and 2017. RS data was obtained from the Copernicus Marine Service (ref
https://marine.copernicus.eu). For SST we used the L4 product
'SEAICE_ARC_PHY_CLIMATE_L4_MY_011_016-TDS', which has spatial resolution of 0.05
degree and daily time resolution. For Chl *a* we used the data service
'OCEANCOLOUR_ARC_CHL_L4_REP_OBSERVATIONS_009_088-TDS' (L4 product
based on the OC5CCI algorithm), which has a spatial resolution of 0.01 degree and monthly time
resolution. Chl *a* concentrations were log-transformed because they span several orders of
magnitude. For both SST and Chl *a* comparisons, the RS data were interpolated to cell center
points of the horizontal FlexSem grid using a bi-linear scheme. Validation was only performed at
spatial points, where RS data has at least one quality-accepted data entry (i.e. sufficient visibility
without ice and cloud cover) for the respective validation periods.
The model skill was assessed by different metrics. The Pearson correlation between observations
and model results was estimated for the seasonal data and spatial data assuming a significance
threshold of $p<0.05$. The other metrics were:
Mean Error (ME) is the mean of the differences between observations *x* and model results *y*:
$$ME = \frac{1}{N} \sum_{i=1}^{N} (y_i - x_i)$$

where *N* is the total number of data points. The Root Mean Square Error (RMSE) is the square
root of the mean squared error between *x* and *y*:
$$RMSE = \sqrt{\frac{1}{N} \sum_{i}^{i=N} (y_i - x)^2}$$

The average cost function (*cf*) is defined as (Radach and Moll 2006):



$$cf = \frac{1}{N}\sum_{i=1}^{N}\frac{|(y_i - x_i)|}{SD(x)}$$

Depending on the *cf* number, it is possible to assess the performance of the model as "very good"
(<1), "good" (1-2), "reasonable" (2-3), and "poor" (>3).
Microzooplankton data was available from the literature for 1996/97 (Levinsen and Nielsen,
2002) and April-May 2011 (Menden-Deuer et al., 2018). Thus, it was not possible to create a
climatology, but the available data was used for visual comparison with model data. Data from
Levinsen and Nielsen (2002) was depth integrated (g-C m$^{-2}$), and converted to mg-C m$^{-3}$ by
assuming that the total biomass was distributed uniformly over the upper 25 m (Levinsen et al.,
2000). Data from Menden-Deuer (2018) was from fluorescence maximum, and this was assumed
to represent the upper 20 m. The conversion from nitrogen to carbon biomass was obtained from
the Redfield ratio=6.625 mol-C mol-N$^{-1}$ and the mol weight of 12 g-C mol$^{-1}$.
**2.9  The impact of sea ice cover and discharge on primary productivity**
An overall indication of the relationship between NPP and sea ice cover and freshwater
discharge was obtained by Pearson product moment correlation analysis between annual
estimates of these for the entire Bay, as defined by the box in figure 1. We further evaluated the
impact of sea ice cover and freshwater discharge on the NPP on a spatial scale. To do this we
perform correlation analysis between the annual NPP and the average sea ice cover March-April
in each model grid cell for 2004-2018. To evaluate the impact of the discharge we performed
similar correlations with average annual surface salinity instead of sea ice cover.  The
assumption behind the choice is that the surface salinity scales with the impact of freshwater
discharge.
To demonstrate the effect of sea ice cover and distance to the glacial outlet on the temporal
development of nitrogen concentration, Chl *a*, and NPP, two stations and two years with
different features were selected. The first station was located in the open bay and the other
station close to the Ilulissat Isfjord (Bay and Glacier station, Fig. 1). The two years 2010 and
2017 were chosen according to differences in both irradiance and sea ice cover, one (2010) with
low sea ice cover and high irradiance and the other (2017) with high sea ice cover and low
irradiance.




To further evaluate the impact of sea ice cover and freshwater discharge we performed some
simple "extreme" model scenarios (Table 1). We tested the potential effect on primary
productivity in 2010 (low sea ice cover) and 2017 (high sea ice cover) in scenarios with no sea
ice, no freshwater discharge or 2 times the reference discharge, as well as the combinations, by
changing the model forcing accordingly.

## 301 3 Results

### 302 3.1 Fresh water discharge and sea ice cover

50 years ago, the average annual liquid runoff from the ice sheet to the study area was generally
~1000 m$^{-3}$ s$^{-1}$ (913±2214 SD m$^{-3}$ s$^{-1}$, 1958-1969), whereas during the last 20 years is has varied
between 2000 and 4500 m$^{-3}$ s$^{-1}$ (2591±724SD m$^{-3}$ s$^{-1}$, 2000-2019) (Fig. 2). The precipitation over
land has also increased from about 200 (197±40 SD m$^{-3}$ s$^{-1}$) to 400-500 m$^{-3}$ s$^{-1}$ (469±77 SD m$^{-3}$ s$^{-1}$
$^{1}$). The calving of solid ice from the glaciers has only been estimated for the last 30 years, but it
also shows an increasing trend although since the maximum in 2013, the production of ice has
been lower (Fig. 2). Thus, for all three sources of freshwater the overall long-term trend is an
increase, but for the model period between 2004 and 2018 no trend was evident (Fig. 3e). The
freshwater discharge from solid ice was relatively constant across the year, whereas the liquid
contribution peaked during summer, from June to August, and drops to almost zero in the winter
(Fig. 3f).
The sea ice cover in Disko Bay has generally decreased during the last 35 years (Fig. 2).
However, the last 15 years have been characterized by large interannual variation with some
years with virtually no ice and others with sea ice cover as in the 1990s. During the model period
the ice generally did not form before late December, and the maximum ice cover was seen in
March (Fig. 3)

### 319 3.2 Validation of the model

The seasonal timing and general level of temperature, salinity, nutrients, Chl *a* and
mesozooplankton agreed well with the data climatology from the field sampling south of Disko
Island (Fig. 4, Table 2). All correlations between observational and model data were significant
(R>0.82). The model performance assessed by the average cost function *cf* was "very good" for
all parameters. Modelled Chl *a* showed highest interannual variability in spring and the





chlorophyll bloom was somewhat too weak (~30% less), and the winter silicate too high, relative
to the climatological mean observations.
The spatial distribution patterns of Chl *a* and temperature at the surface were compared to
satellite estimates for the two years 2010 and 2017 used in the scenarios representing low and
high sea ice cover, respectively (Table 3, Fig. C1). The correlations were significant for all
relations ($p<0.01$), and the *cf* number was "very good" or "good" for all (Table 3). Surface
temperature tended to be higher in spring and lower in summer in the model compared to the
satellite estimates. Chl *a* concentrations were generally higher in the model than in the satellite
data, especially in spring 2017 (Fig. C1).

### 3.3    Seasonal and spatial patterns of NPP in Disko Bay

Primary production starts as sea ice cover decreases and irradiance increases in February (Fig. 3).
Extensive sea cover may reduce light availability in the water column and thereby limit
production, and the interannual variation in NPP is highest in April because of the variation in
sea ice cover, causing light availability in the water to vary accordingly. Highest NPP was in
May and June with about 800 mg-C m$^{-3}$ d$^{-1}$ when light influx was highest and sea ice was
entirely melted (Fig. 3).
The impact of sea ice is illustrated by comparing a year with low (2010) and high (2017) sea ice
cover, where the spring bloom is about 25-30 days earlier in 2010 than in 2017 (Fig. 5).
Comparing a station close to and far from the glacier illustrates the potential impact of the fresh
water peak in late summer, as NPP is 2-3 times higher during this period at the station close to
the glacier (Fig. 5).
Concerning the spatial distribution in the spring period (March to June), high NPP was seen
across the bay, with the lowest values found southeast of the Disko Island and southwest of the
Bay following the bathymetry. In the later summer period (July to October), primary production
was more confined to the coast (Fig. 6).

### 3.4    Annual variability of NPP

The annual average NPP in the Bay estimated from the model varied between 90 and 147 g-C
m$^{-2}$ year$^{-1}$ with an average of 129±16 (SD) (Fig. 3). Generally, years with high sea ice cover in
spring had lower average annual NPP (Fig. 3, Pearson product moment correlation coefficient *r*





= -0.63, p=0.01), while higher discharge was associated with higher annual primary productivity
(Fig. 3, $r$ = 0.51, p=0.05).
To evaluate the spatial dependency, we performed an analysis of the correlation between the sea
ice cover in March to April and the annual NPP in each model grid cell. This showed a negative
relationship widespread in the model domain, i.e. the more sea ice, the lower NPP (Fig. 7). One
exception was in the south part of the model domain, where the correlation was positive. The
impact of the freshwater discharge on the NPP was generally positive in areas up to ~50 km from
the discharge and additionally in the northern part of Disko Bay, as reflected by the negative
correlation to surface salinity in these areas (Fig. 7).

### 363 **3.5 Model scenarios with sea ice cover and discharge**

We studied some simple model scenarios where sea ice cover was assumed to be zero and the
discharge was either doubled or cut off, with basis in 2010 and 2017, which had low and high sea
ice cover, respectively, and opposite discharge (Fig. 3). These scenarios underline the
complexity of the dynamics of the system, with some areas experiencing increased NPP while
others experience a decrease (Figs. 8, 9). Furthermore, it allows us to evaluate the impact of the
uncertainty of actual freshwater runoff. The year 2017 had relatively high and late ice cover (Fig.
3) and applying a scenario of no ice leads to an increase in bay-scale annual NPP of 34 %,
although spatial variability is high and annual NPP changes vary between -20% and 98% (Fig.
9). For 2010, a year that already had low sea ice cover, the same scenario led to minor changes in
the annual NPP on bay scale (2 %, Fig. 8). For both years, the omission of freshwater discharge
generally led to a decrease in annual NPP; this effect was small on the bay scale (-2 to 0%), but
reached -64% in near-coastal areas under glacial/runoff influence. Similarly, the effect of
doubling of the discharge was minor on the bay scale (0-1%), but reached up to 55 and 68 %
NPP increase in runoff-influenced areas in 2010 and 2017, respectively. The effects of sea ice
and freshwater discharge changes combined in an approximately additive manner (Figs. 8, 9).
When the forcing from sea ice cover and freshwater discharge were set to be zero in 2010 and
2017, NPP in 2017 was were still 20% smaller than the 2010. This illustrates the importance of
other factors for NPP like wind, cloud cover and inflow to the bay.



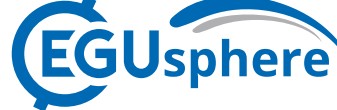

## 4   Discussion

Primary productivity is an essential ecosystem service that shapes the structure of the marine
ecosystem and fuels higher trophic levels such as fish that is vital for the Greenlandic society. It
is therefore important to estimate potential outcomes for primary production under the continued
warming and subsequent ice melt. For the coastal ocean, especially around Greenland, it is
imperative to quantify how changes in sea ice cover and run-off combine to determine the
availability of the two key resources, light and nitrate, determining the magnitude and phenology
of primary production. Sea ice cover and run-off influence light and nitrate availability through
several intermediate processes and their peak impact often occurs in different areas and in
different months. The spatial-temporal variability and complexity of processes involved requires
an approach where detailed *in situ* observations are combined with remote sensing and
modelling. The present study is to our knowledge the first attempt to apply this approach for
coastal Greenland.

Our model results show that reduction in spring sea ice cover changes the plankton phenology
but also increases the magnitude of annual production in Disko Bay. This suggests that there is a
replenishment of nitrate into the photic zone to sustain the continued productivity beyond the
initial depletion following the spring bloom. Part of the nitrate input is coupled to the run-off, but
the high modelled productivity from April to July, when liquid run-off is limited suggest that
vertical mixing fueled by wind and tide is important. That less sea ice cover will lead to
increased NPP is in agreement with other studies from the open Arctic areas (Arrigo and van
Dijken, 2015; Vernet et al., 2021). In other Greenland fjords, the turbulence driving vertical
mixing has been shown to be very low (Bendtsen et al., 2021; Randelhoff et al., 2020), but is
seems likely that the open Disko Bay with a tidal amplitude of up to 3 m (Thyrring et al., 2021)
could have an efficient vertical flux of nitrate into the photic zone.

Our study site was chosen because the Disko Bay in mid-west Greenland is considered a hot-spot
for marine biodiversity and fisheries, and because it is an area where both sea ice cover and
glacial run-off are likely to be important for productivity. But regional variability is high across
the coastal ocean around Greenland. For example, ice cover is very limited in most of SW
Greenland and is unlikely to drive changes in future primary production, whereas glacial run-off
is less in NE Greenland compared to the rest of Greenland. Furthermore, the dominance of land





or marine terminating glaciers as in Disko Bay will be important for the outcome of increased
glacial run-off on individual fjord scale (Hopwood et al., 2020; Lydersen et al., 2014). Finally,
winter concentration of nitrate and vertical gradients in summer differ between the East and West
coast, with low nitrate content in the East Greenland Current generally causing lower
productivity compared to West Greenland (Vernet et al. 2021).

### 4.1 Phenology of primary producers

A main advantage of the model is that it allows us to estimate the productivity with a higher
temporal and spatial resolution than would be possible from measurements alone. The sea ice
cover had a clear effect on the spring NPP. When sea ice cover is low, spring NPP is starting
earlier compared to years with high sea ice cover, and the largest variation in NPP between years
is seen in the spring months (Fig. 3).  The performed scenarios support the importance of sea ice
cover, i.e. the absence of sea ice leads to a considerable increase in the annual NPP on bay scale
(Fig. 9).  Potentially, NPP could start as early as February if considering the light availability.
However, for NPP to increase would also require the water column to stabilize, i.e. wind mixing
would need to be sufficiently low (Tremblay et al., 2015). In contrast, the timing of the formation
of the sea ice in fall is not important for the primary productivity, since the sea ice in Disko Bay
does not form before the light has largely disappeared. This is in contrast to high Arctic systems
where sea ice normally forms earlier and a delay in the formation of sea ice in fall may result in
autumn blooms (Ardyna et al., 2014).

### 4.2 Spatial distribution of NPP

In our analysis, we see a positive effect of the freshwater discharge on the primary productivity
locally and during the summer months. This effect is related to the upwelling that is enhanced by
the freshwater discharge (Fig. C2, C3). The nutrient concentration in the discharge (1.25 µM,
Hopwood et al., 2020) is lower than the average concentration in the upper 30 m during summer
at the station near the glacier (e.g. ~4 µM $NO_3$)  (Fig. 7), and will therefore not lead to increased
NPP. This is in accordance with the general picture from glacial affected environments. River
discharge may on the other hand carry higher nutrient concentrations, particularly of nitrogen
(Hopwood et al., 2019).
We used two approaches to evaluate the spatial scale of the effect freshwater discharge. The
correlation analyses using salinity as a proxy for the discharge (Fig. 7) suggest that the discharge



may influence ~50 km away from the source.  The scenarios where we alter the discharge
suggest that the effect is only a couple of percent considering NPP on the Bay scale, whereas on
a more local scale near the glacier the importance is higher (-64% to 147%, Fig. 8 and 9). In the
Godthåbsfjord, which is situated further south at the west coast of Greenland it was found that 1-
11% of the NPP in the Fjord systems is supported by entrainment of N by the three marine
terminating glaciers (Meire et al., 2017). However, considering only the parts of the fjord
directly impacted by the discharge the estimate were 3 times higher (Hopwood et al., 2020).
Analyses from Svalbard fjords impacted showed positive spatiotemporal associations of
chlorophyll a with glacier runoff for 7 out of 14 primary hydrological regions but only within 10
km distance from the shore (Dunse et al., 2022).
The modelling in this study allows us to evaluate the combined effect of changes in sea cover
and freshwater discharge in the coastal ecosystem of the Disko Bay. Importantly, this study also
illustrates that within the Arctic coastal zone, the combination of different climate change effects
may lead to different responses within relatively small distances. Thus, while we can suggest a
general increasing trend in the NPP, this may not be evident when considering local
observations. This is important to consider when planning and evaluating field investigations.

### 4.3   Modelled NPP versus other estimates

The biogeochemical model was validated using all available observations. These are all
concentrations (nutrients) or standing stocks (phytoplankton, zooplankton). The satisfactory
validation is an indication that the rates are also adequately described.  Still, it is desirable also to
have direct comparison with rate measurements. There are no available NPP measurements for
our modelling period. However, data are available from 1973-1975 (Andersen, 1981) and
1996/97 (Levinsen and Nielsen, 2002) and 2003 (Sejr et al., 2007). The data from 1996/97 were
*in situ* bottle incubations in the upper 30 m, and no further information on methodology was
given (referred to as unpublished). The sea ice cover was generally high in Disko Bay at that
time (Fig. 4) and we therefore compare the seasonal development to our model estimates from
2017, a year with extensive sea ice cover. The estimate of the annual production from 1996/97
was 28 gC $m^{-2}$ $d^{-1}$ less than half the estimate from 1970s of 70 gC $m^{-2}$ $d^{-1}$, and the modeling
estimates from 2017 of 82 gC $m^{-2}$ $d^{-1}$ at the same station. The measurements do, however, both
agree with the model on the seasonal timing of NPP with an increase in NPP between March and





April, and the Pearson correlation coefficients between measurements and model results were
0.84, p<0.001 (1996/7) and 0.69, p<0.05 (1973-75). Data from 2003 (Sejr et al., 2007) are from a
shallow cove only in two shorter periods, but the production of 195 mgC m$^{-2}$ d$^{-1}$ in April aligns
well with our estimates, whereas the value in September 27 mgC m$^{-2}$ d$^{-1}$ is somewhat lower.
Average estimates of NPP from Arctic glacial fjords with marine terminating glaciers are
reported to be 400-800 mg-C m$^{-2}$ d$^{-1}$ during July to September (Hopwood et al., 2020). In the
Arctic Ocean, shelf regions estimates from satellite observations are 400-1400 mgC m$^{-2}$ d$^{-1}$ in
April to September during 1998 to 2006 (Pabi et al., 2008). Thus, overall, our model estimates of
NPP in Disko Bay of 378-815 mgC m$^{-2}$ d$^{-1}$ between April and September (Fig. 3) are in the same
range as other estimates.
In another modelling study, a physically-biologically coupled, regional 3D ocean model
(SINMOD) was compared with ocean color remote sensing (OCRS). Both OCRS and SINMOD
provided similar estimates of the timing and rates of productivity in of the shelves around
Greenland (Vernet et al., 2021). In the region including Disko Bay, the modelled NPP was
generally suggested to be much lower (20-23 gC m$^{-2}$ yr$^{-1}$) than our estimate (90-147 gC m$^{-2}$ yr$^{-1}$)
and the bloom was suggested to generally start later (late May). However, their model mainly
covered the shelf area north of Disko Bay and did not resolve the plume outside the ice fjord.
Moreover, the estimates from OCRS (50 gC m$^{-2}$ yr$^{-1}$) were about double the modelled values,
and furthermore could only be recorded after ice break-up when the bloom was already on its
maximum (Vernet et al., 2021), suggesting that it could be much higher.
**4.4   Uncertainty and potential model improvement**
We model the impact of turbidity on light conditions in the water column as a simple relationship
between salinity and light attenuation. More sophisticated light models may be applied in future
models (Murray et al., 2015). However, in a relatively open water system like Disko Bay, the
effect of increased light attenuation due to increased turbidity is only expected within 5-10
kilometers of the glacial outlet. Moreover, we do not expect an impact on the total NPP in the
bay since the nutrients will anyway be used within the bay. A comparison between the spatial
distribution of surface Chl a assessed by satellite and the model showed a significant correlation
and the model performance were evaluated good to excellent (Table 3). Still, visual inspections
of the two maps suggest that the effect of the discharge on the Chl *a* spatial distribution were





more local and concentrated in the model than what is suggested by the satellite estimates (Fig.
C1). Thus, a higher precision in the spatial distribution of the phytoplankton may be achieved by
improving the model parametrization of light attenuation, e.g. by inserting a passive tracer
reflecting the turbidity in melt water.
The uncertainty in the different fresh water discharge source may impact our estimates of marine
productivity differently. Liquid runoff uncertainty and errors are more likely to be random than
bias, and when averaged together (over large spatial areas or times) the uncertainty is reduced
(Mankoff et al., 2020b). Conversely, solid ice discharge uncertainty is comes primarily from
unknown ice thickness, which is time-invariant and therefore must be treated as a bias term
(Mankoff et al., 2020a). It does not reduce when averaged in space or time.
We do not specifically model the subglacial discharge of freshwater from the marine terminating
glaciers or from the numerous large icebergs in the bay. Instead, the freshwater discharge was
distributed equally across the upper 100 m in the locations where marine terminating glaciers
were present. Thus, our model is not currently able to resolve the small-scale mixing between
sub-glacial discharge and ambient fjord water in the plume directly in front of the glacier. A
study from another Greenland fjord suggests efficient mixing near the glacial terminus, which
means that the freshwater fraction in the surface water near the glacial front is only 5-7%, which
indicates that the mixing ratio between sub-glacial discharge and fjord water is 1 liter of
meltwater to 13-16 liters of fjord water (Mortensen et al., 2020). The capacity of buoyancy
driven upwelling of subglacial discharge to supply nutrients to the photic zone depends on
several factors including the depth of the freshwater input and the density and nutrient content of
the ambient fjord water. Our approach to distribute the freshwater input in the upper 100 m is a
first attempt to simulate the average conditions across the study area. We were able to reproduce
the general pattern of upwelling (Fig C2+C3) and spatial dynamics of productivity, but the
magnitude could be underestimated. Models of high spatial and process resolution are mainly
developed to describe the transports of heat and salt to glacial ice, in order to estimate the melt
(Burchard et al., 2022). If the focus is to describe the fine scale processes in front of the glacier,
the development within these models may in the future be implemented in ocean models.



## 4.5 Conclusions

Two important drivers of changes in the Arctic coastal ecosystems are sea ice cover and glacial freshwater discharge. This modelling study estimates the response of the pelagic net primary (NPP) production to changes in sea ice cover and freshwater run-off in Disko Bay, West Grenland, from 2004 to 2018. The difference in annual production between the year with lowest and highest annual NPP was 63%. Our analysis suggests that sea ice cover was the more important of the two drivers of annual NPP through its effect on spring timing and annual production. Fresh water discharge, on the other hand, had a strong impact on the summer NPP near to the glacial outlet. Hence decreasing ice cover and more discharge can work synergistically and increase productivity of the coastal ocean around Greenland.

## 5 Author contribution

EFM, MAM, MS conceptualized the study. MAM, JL, EFM was responsible for the FLEXSEM development and validation, MHR for HYCOM-CICE, PW for the Arctic 'A20' model, KM for MAR/ RACMO, and AC for the remote sensing data. MAM and EFM analyzed, synthesized and visualized the data. EFM prepared the initial draft, and all authors contributed to review and editing.

## 6 Competing interests

The authors declare that they have no conflict of interest.

## 7 Acknowledgements

This research has been supported by the Programme for Monitoring of the Greenland Ice Sheet (PROMICE) and the European Union's Horizon 2020 research and innovation program (INTAROS, grant no. 727890), and the Danish Environmental Protection Agency (MST-113 00095 and j-nr 2019 - 8443). MHR was funded by the Danish State through the National Centre for Climate Research. PW was funded by the Joint Programming Initiative Healthy and Productive Seas and Oceans (JPI Oceans) project CE2COAST and the EU Horizons 2020 project FutureMARES, and used resources provided by the Norwegian Metacenter for Computational Science and Storage Infrastructure (Notur/Norstore projects nn9490k, nn9630k, and ns9630k). Data from the Greenland Ecosystem Monitoring Programme were provided by the Department





of Bioscience, Aarhus University, Denmark, in collaboration with the Department of
Geosciences and Natural Resource Management, Copenhagen University, Denmark. The authors
are solely responsible for all results and conclusions presented, and they do not necessary reflect
the position of the Danish Ministry of the Environment or the Greenland Government.



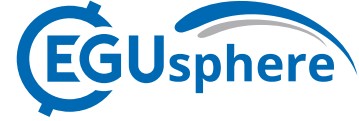

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




## 8  Tables

Table 1: Characteristics of the reference model runs of 2010 and 2017, and the annual average
NPP in the bay obtained from scenarios runs with changes in the sea ice cover and the freshwater
discharge (Figure 8 and 9). SD are the standard variation between the different model grid cells.

|  |  |  |  | 2010 | | 2017 | |
|---|---|---|---|---|---|---|---|
| **Reference** | Average annual primary production | gC m⁻² yr⁻¹ | | 147 | ±41 | 90 | ±28 |
| | Average annual discharge | m³ s⁻¹ | | 6275 | | 4058 | |
| | Average annual sea ice cover, March-April | % | | 24 | | 79 | |
| **Scenarios** | Average annual primary production | gC m⁻² yr⁻¹ | No sea ice | 150 | ±50 | 120 | ±35 |
| | | | No freshwater discharge | 144 | ±53 | 90 | ±46 |
| | | | No sea ice, No freshwater discharge | 147 | ±47 | 119 | ±32 |
| | | | 2 x freshwater discharge | 149 | ±48 | 90 | ±45 |
| | | | No sea ice, 2 x freshwater discharge | 152 | ±53 | 122 | ±35 |




Table 2: Statistics for seasonal comparison between observational data (monthly climatology)
and model data (monthly average from 2005 to 2018) at the Disko Bay Station. *N*=12 for
copepods, *N*=11 for temperature, salinity and Chl *a* and *N*=10 for other variables (see Figure 4).
All correlations were significant (*p*<0.01).

| | Unit | Model error | RMSE | Correlation | *cf* |
|---|---|---|---|---|---|
| Temperature | °C | -0.28 | 0.96 | 0.94 | 0.31 |
| Salinity | - | -0.09 | 0.21 | 0.79 | 0.56 |
| NO$_3$ | mmol m$^{-3}$ | 0.00 | 1.43 | 0.87 | 0.39 |
| Silicate | mmol m$^{-3}$ | 0.78 | 1.70 | 0.83 | 0.66 |
| Phosphate | mmol m$^{-3}$ | -0.01 | 0.12 | 0.82 | 0.46 |
| *Chl a* | mg m$^{-3}$ | 0.03 | 0.97 | 0.87 | 0.37 |
| Copepod biomass | mgC m$^{-3}$ | 0.83 | 4.66 | 0.94 | 0.23 |






Table 3: Statistics for the spatial comparison between remote sensing data and surface model
data for spring (April-June) and summer (July-September) in 2010 and 2017. In spring 2017,
only June is included due to ice cover in April-May. *N*=6145, and all correlations were
significant (*p*<0.01).

|  | Model error | RMSE | Correlation | *cf* |
|---|---|---|---|---|
| *Surface temperature* |  |  |  |  |
| 2010 spring | 0.8 | 1.3 | 0.45 | 1.0 |
| 2010 summer | -1.4 | 2.0 | 0.14 | 1.5 |
| 2017 spring | 0.8 | 1.4 | 0.58 | 0.9 |
| 2017 summer | -2.0 | 2.3 | 0.33 | 0.2 |
| *$Log_{10}$ (Chl a [mg/m$^3$])* |  |  |  |  |
| 2010 spring | 0.6 | 0.7 | 0.30 | 0.4 |
| 2010 summer | 0.5 | 0.8 | 0.33 | 0.2 |
| 2017 spring | 1.7 | 1.8 | 0.29 | 1.7 |
| 2017 summer | 0.9 | 1.1 | 0.46 | 1.2 |




## 9   Figures

Figure 1:  Map of Disko Bay with the bathymetry, the Flexsem model grid, position of fresh water sources (red dots: land runoff, red dots with black circle: land + ice runoff), position of two stations presented in more detail, and the area used for calculation of the average Disko Bay primary production (red box).

Figure 2:  Development in freshwater discharge and sea ice cover over time. a) Fresh water discharge from the Greenland ice sheet divided into liquid from precipitation over land (Land runoff), liquid deriving from melt from the Greenland Ice sheet/glaciers (Ice runoff) and ice deriving directly from the glacier (solid ice) 1960 to 2019, and b) number of days with more than 40% sea ice cover from 1986 to 2019, derived from satellite measurement (AICE), by the sea ice model providing input to the this study (CICE), and by visual observation at Arctic Station, Qeqertarsuaq (AS).

Figure 3:  Primary production, sea ice cover and freshwater discharge in Disko Bay from 2004 to 2018. Primary production and sea ice cover are assessed in the red square in Fig 1, whereas the freshwater discharge are from the full model domain. (a) Average annual primary production (gC $m^{-2}$ $year^{-1}$)± SD (variation between model grid cells), (b) the average monthly primary production (mgC m-2 day-1) ± SD (variation between years), light is average from Arctic station (2010-2019), (c) the annual average sea ice cover in March and April (%), (d) the average monthly sea ice cover (%), (e) the average annual fresh water discharge ($m^3$ $s^{-1}$), and (f) the average monthly fresh water discharge (1000 $m^3$ $s^{-1}$).

Figure 4: Comparison of monthly means (±SD) of observations and model data (2004-2018) at 69°14'N, 53°23'W for (a) temperature (℃), (b) salinity, (c) nitrate (mmol $m^{-3}$), (d) silicate (mmol $m^{-3}$), (e) phosphate (mmol $m^{-3}$), (f) Chl $a$, (mg $m^{-3}$), (g) microzooplankton biomass (mgC $m^{-3}$), and (h) mesozooplankton biomass (mgC $m^{-3}$). Means are averaged over 0-20 m depth, except for mesozooplankton which it is 0-50 m.

Figure 5: Sea ice cover (%), average nitrate concentration in 0-30 m (mmol $m^{-3}$) average Chl $a$ concentration in 0-30 m (mg $m^{-3}$) and primary production (mgC $m^{-2}$ $d^{-1}$) at a station in open Bay (Bay Station) and at one close to the glacier (Glacier Station) (Fig. 1) in 2010 and 2017.





Figure 6:  Average spatial distribution of primary production (gC m$^{-2}$) in 2010 and 2017
respectively for the periods A)+D) March-October, B)+E) March-June and C) +F) July-October.
Figure 7: Correlation coefficients between the annual primary production (a) and average sea ice
cover in March-April and (b) and surface salinity across the period 2004-2018.
Figure 8: Response of the annual primary production to simple scenarios of changes in sea ice
cover and freshwater discharge (Q) in 2010 expressed as percentage change relative to the
standard model run. The percentages in the bottom of the figure are the changes in primary
production in the total area shown. The following model scenarios were run (Table 1): (a)
standard model run, (b) assuming no sea ice cover, (c) assuming no freshwater discharge from
the Greenland ice sheet,  (d) the combination of (b) and (c), (e) assuming 2 times the freshwater
discharge of the standard run, and (f) the combination of (b) and (e).
Figure 9: Response of the annual primary production to simple scenarios of changes in sea ice
cover and freshwater discharge (Q) in 2017 expressed as percentage change relative to the
standard model run. The percentages in the bottom of the figure are the changes in primary
production in the total area shown. The following model scenarios were run (Table 1): (a)
standard model run, (b) assuming no sea ice cover, (c) assuming no freshwater discharge from
the Greenland ice sheet, (d) the combination of (b) and (c), (e) assuming 2 times the freshwater
discharge of the standard run, and (f) the combination of (b) and (e).





Figure 1: Map of Disko Bay with the bathymetry, the Flexsem model grid, position of fresh water sources (red dots: land runoff, red dots with black circle: land + ice runoff), position of two stations presented in more detail, and the area used for calculation of the average Disko Bay primary production (red box).

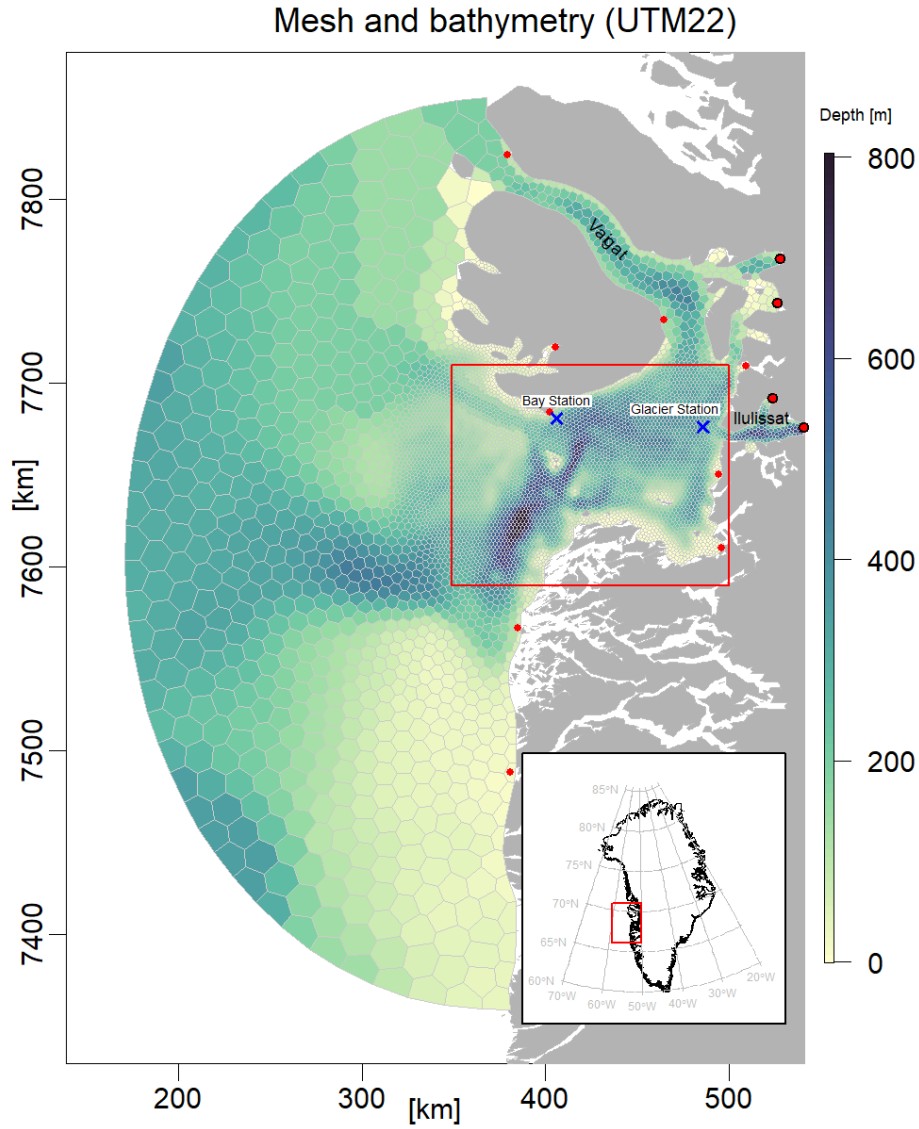






Figure 2: Development in freshwater discharge and sea ice cover over time. a) Fresh water discharge from the Greenland ice sheet divided into liquid from precipitation over land (Land runoff), liquid deriving from melt from the Greenland Ice sheet/glaciers (Ice runoff) and ice deriving directly from the glacier (solid ice) 1960 to 2019, and b) number of days with more than 40% sea ice cover from 1986 to 2019, derived from satellite measurement (AICE), by the sea ice model providing input to the this study (CICE), and by visual observation at Arctic Station, Qeqertarsuaq (AS).

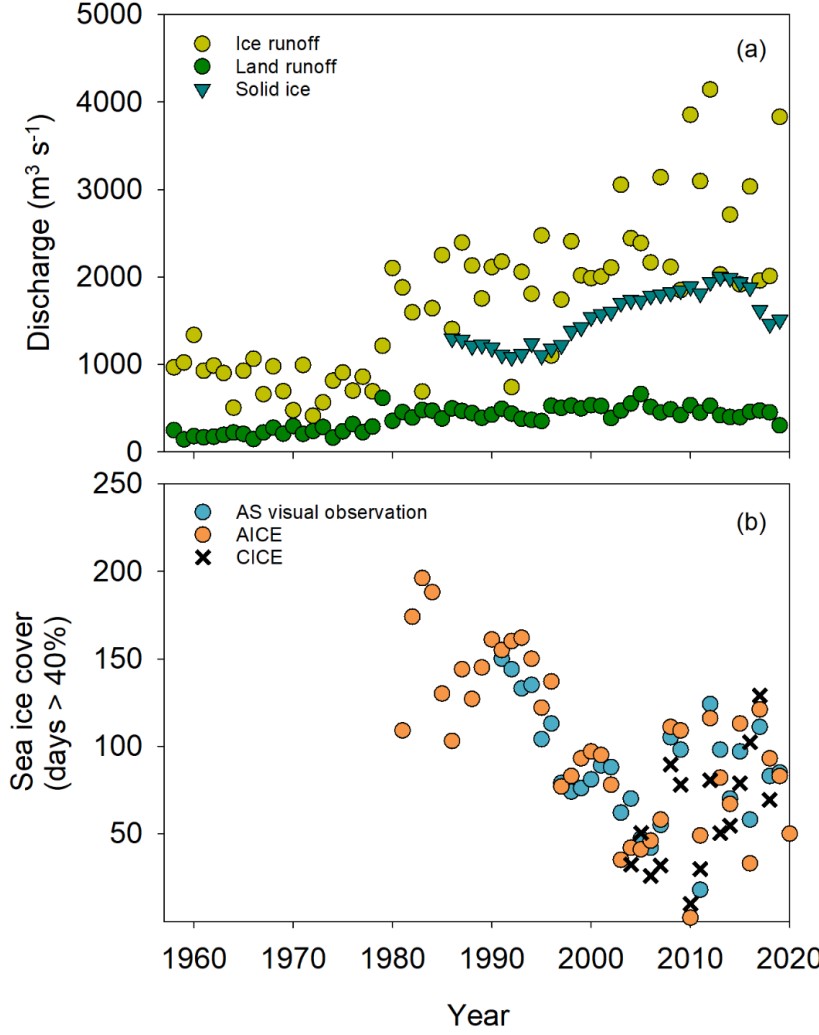




Figure 3: Primary production, sea ice cover and freshwater discharge in Disko Bay from 2004 to 2018. Primary production and sea ice cover are assessed in the red square in Fig 1, whereas the freshwater discharge are from the full model domain. (a) Average annual primary production (gC m$^{-2}$ year$^{-1}$)± SD (variation between model grid cells), (b) the average monthly primary production (mgC m-2 day-1) ± SD (variation between years), light is average from Arctic station (2010-2019), (c) the annual average sea ice cover in March and April (%), (d) the average monthly sea ice cover (%), (e) the average annual fresh water discharge (m$^3$ s$^{-1}$), and (f) the average monthly fresh water discharge (1000 m$^3$ s$^{-1}$).

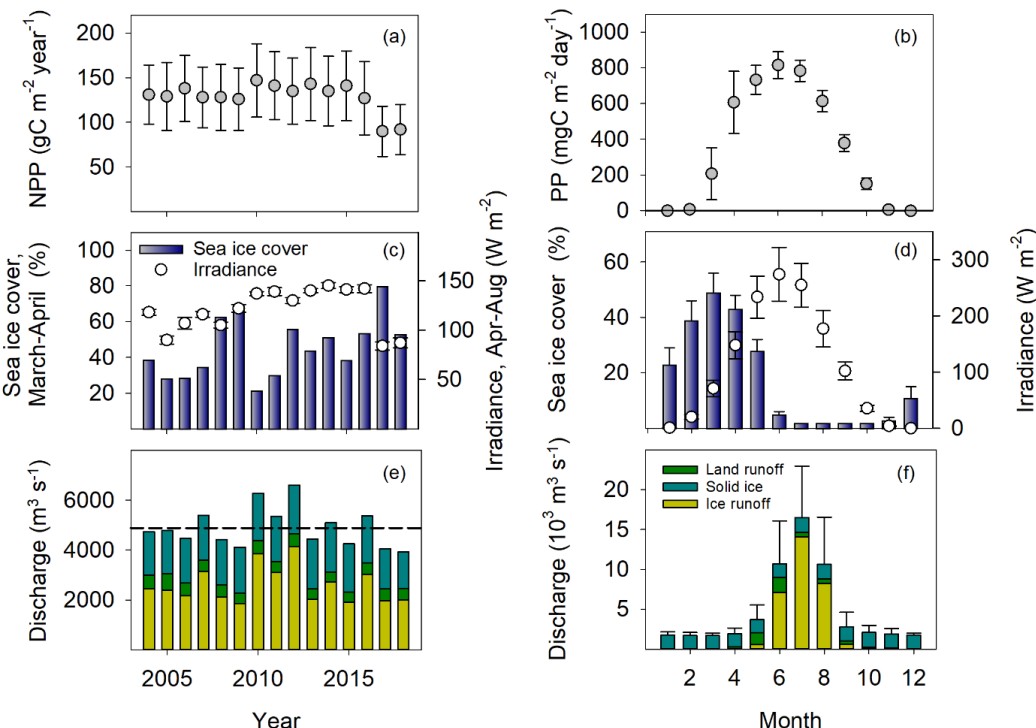






Figure 4: Comparison of monthly means (±SD) of observations and model data (2004-2018) at 69°14'N, 53°23'W for (a) temperature (°C), (b) salinity, (c) nitrate (mmol m$^{-3}$), (d) silicate (mmol m$^{-3}$), (e) phosphate (mmol m$^{-3}$), (f) Chl *a*, (mg m$^{-3}$), (g) microzooplankton biomass (mgC m$^{-3}$), and (h) mesozooplankton biomass (mgC m$^{-3}$). Means are averaged over 0-20 m depth, except for mesozooplankton which it is 0-50 m.

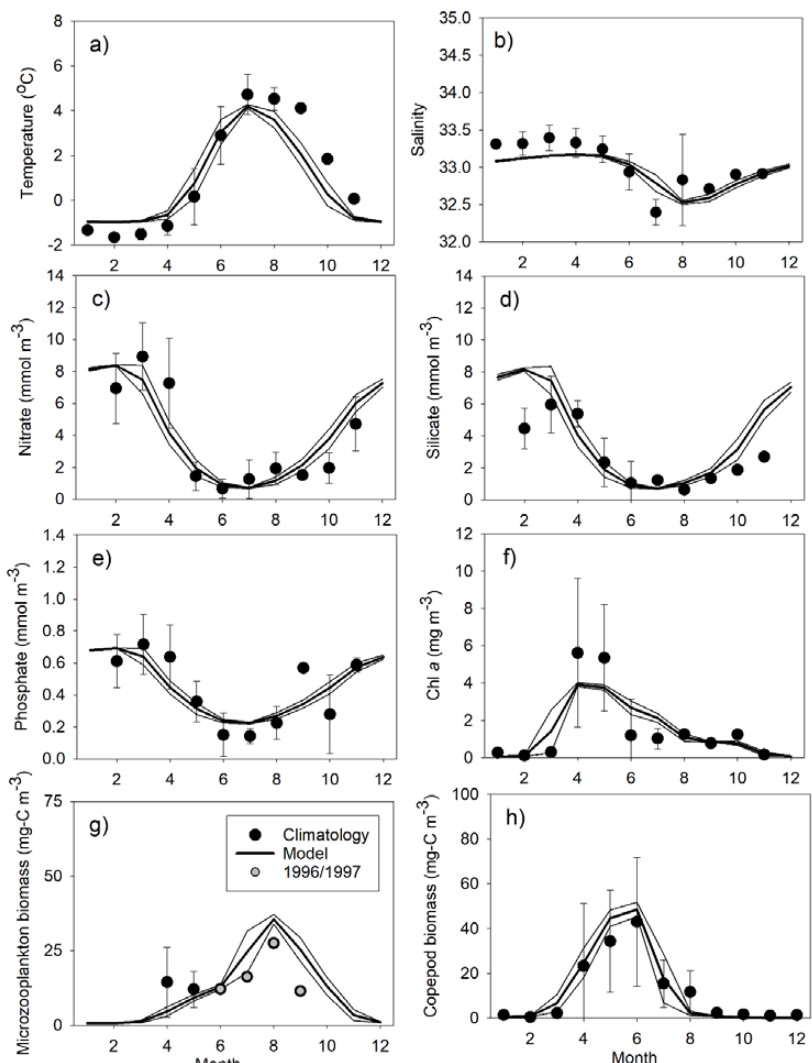








Fig 5: Sea ice cover (%), average nitrate concentration in 0-30 m (mmol m$^{-3}$) average Chl *a* concentration in 0-30 m (mg m$^{-3}$) and primary production (mgC m$^{-2}$ d$^{-1}$) at a station in open Bay (Bay Station) and at one close to the glacier (Glacier Station) (Fig. 1) in 2010 and 2017.

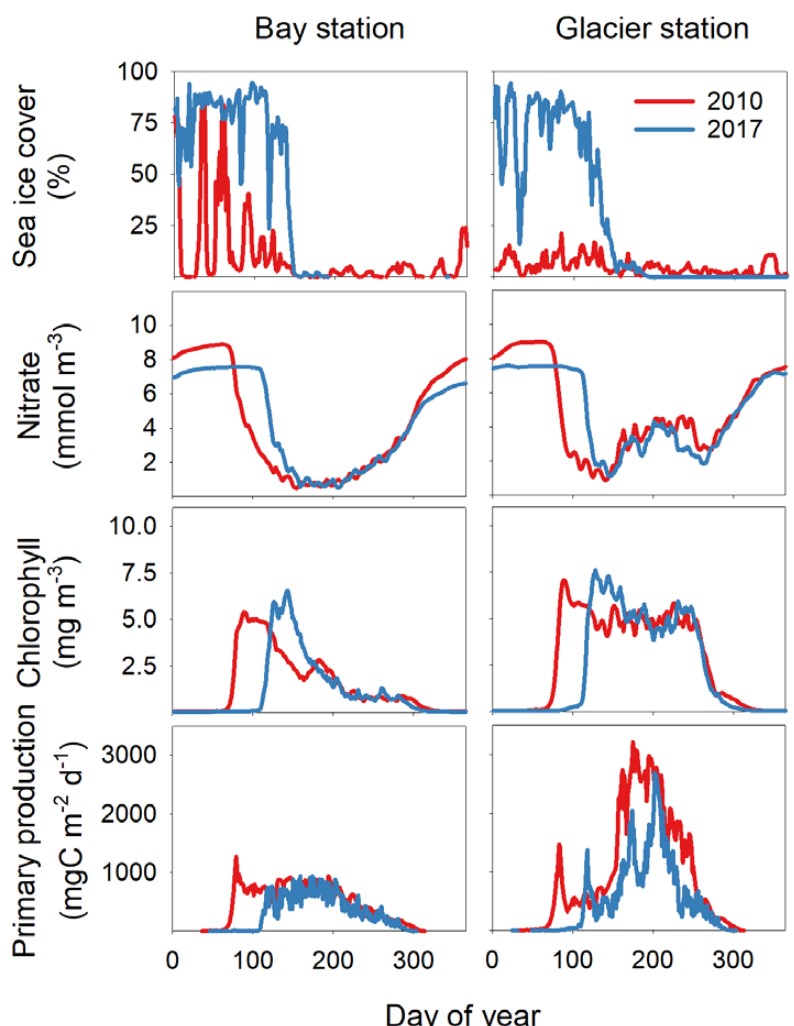



Fig 6: Average spatial distribution of primary production (gC m$^{-2}$) in 2010 and 2017 respectively for the periods A)+D) March-October, B)+E) March-June and C) +F) July-October.

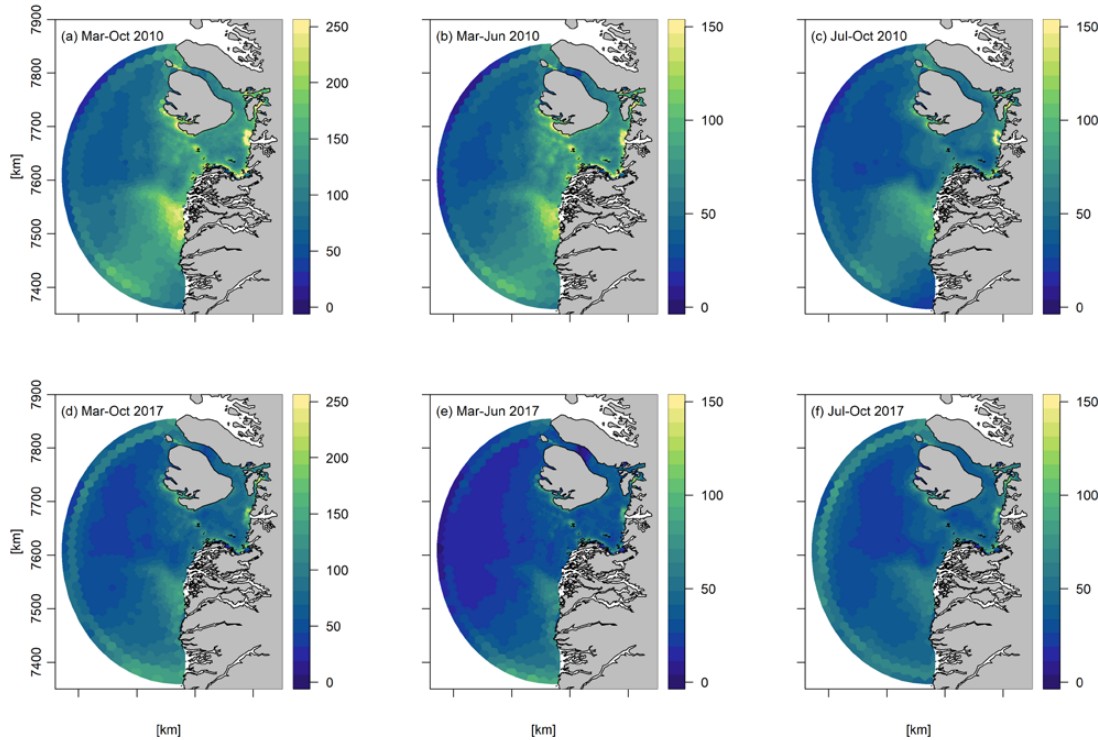



Frig 7: Correlation coefficients between the annual primary production (a) and average sea ice cover in March-April and (b) and surface salinity across the period 2004-2018.

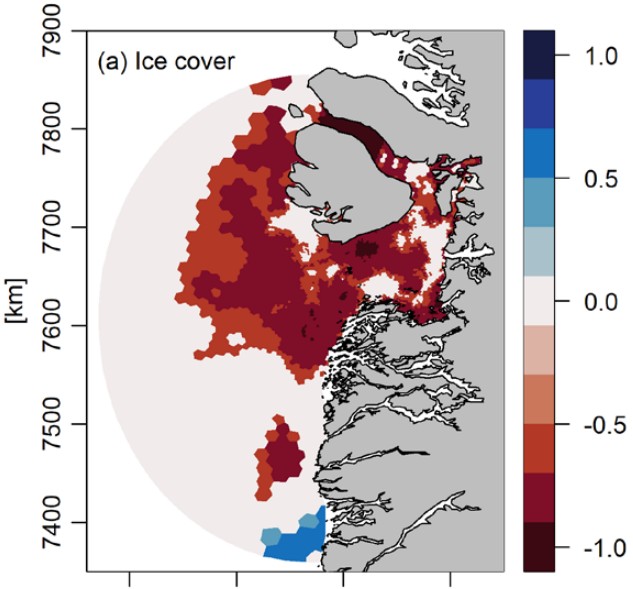

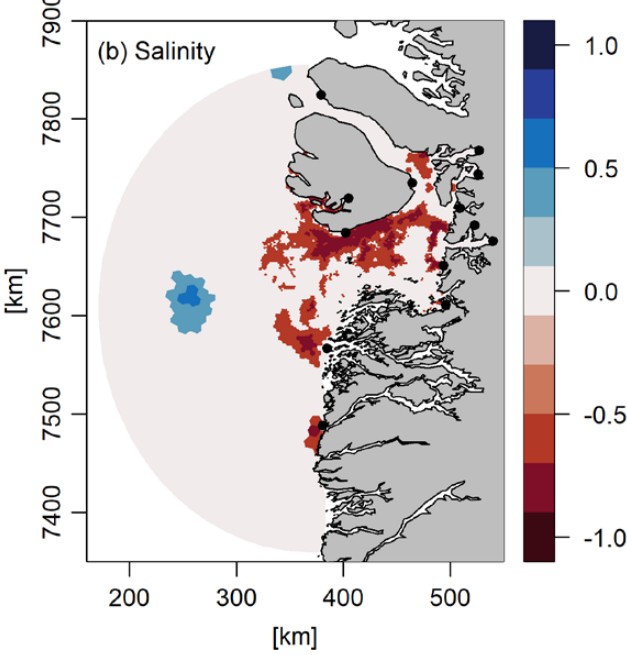




Fig 8: Response of the annual primary production to simple scenarios of changes in sea ice cover and freshwater discharge (Q) in 2010 expressed as percentage change relative to the standard model run. The percentages in the bottom of the figure are the changes in primary production in the total area shown. The following model scenarios were run (Table 1): (a) standard model run, (b) assuming no sea ice cover, (c) assuming no freshwater discharge from the Greenland ice sheet, (d) the combination of (b) and (c), (e) assuming 2 times the freshwater discharge of the standard run, and (f) the combination of (b) and (e).

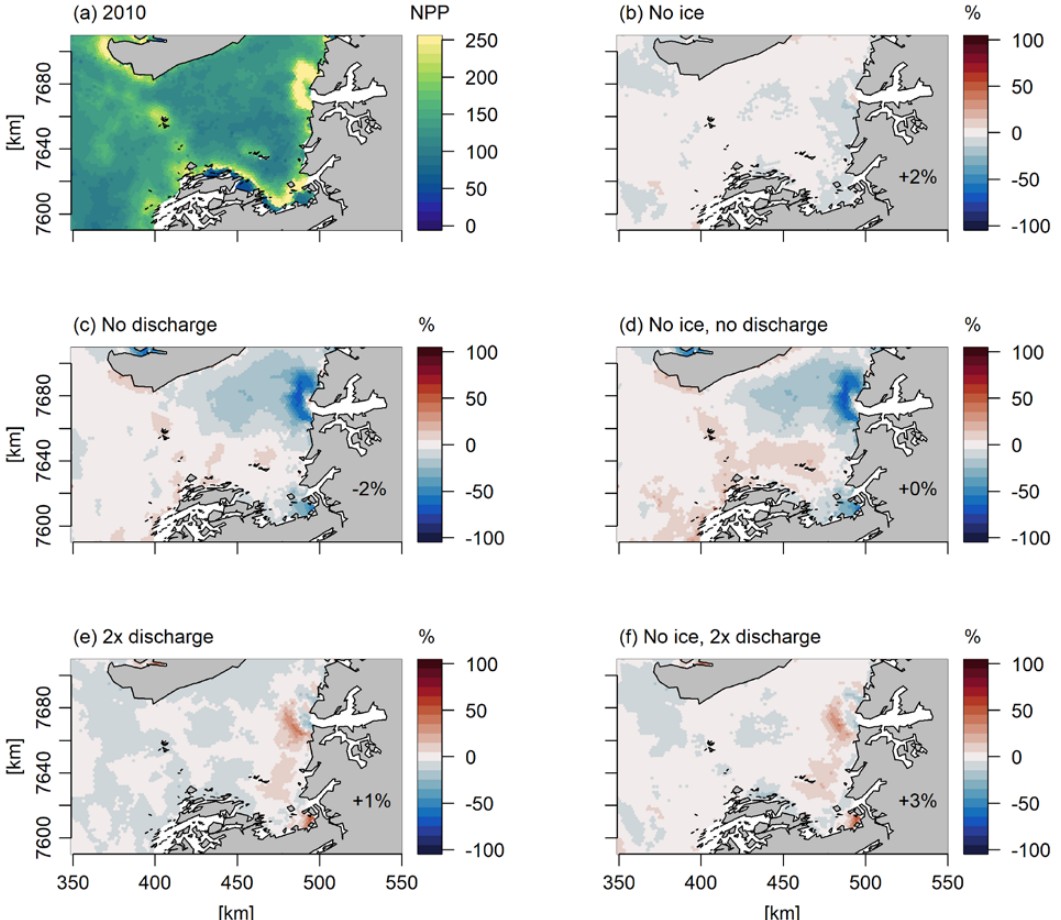




Fig 9: Response of the annual primary production to simple scenarios of changes in sea ice cover and freshwater discharge (Q) in 2017 expressed as percentage change relative to the standard model run. The percentages in the bottom of the figure are the changes in primary production in the total area shown. The following model scenarios were run (Table 1): (a) standard model run, (b) assuming no sea ice cover, (c) assuming no freshwater discharge from the Greenland ice sheet, (d) the combination of (b) and (c), (e) assuming 2 times the freshwater discharge of the standard run, and (f) the combination of (b) and (e).

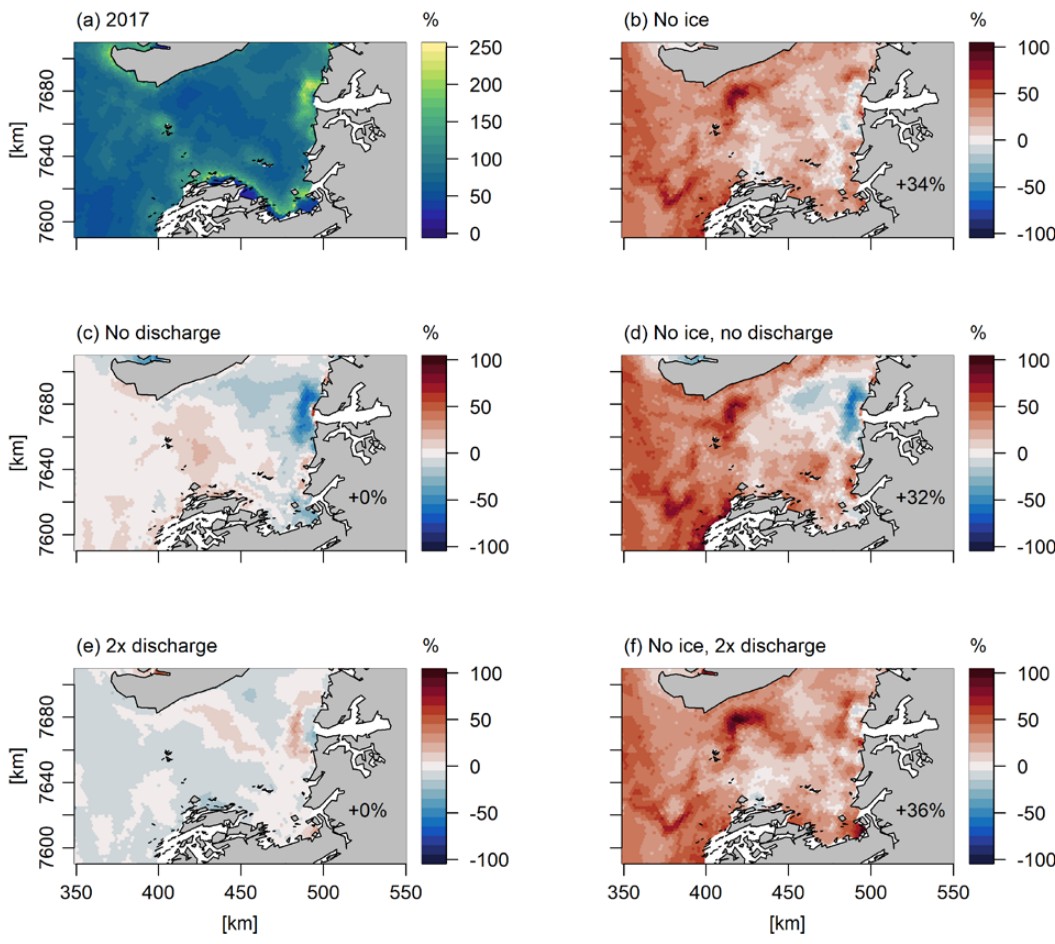




## 10 Appendices

### 10.1 Appendix A, Ecological model constants

Table A.1. Constants in the FlexSem ecological Disko Bay model.

| Parameter | Description | Numerical value | Units |
|---|---|---|---|
| **Phytoplankton** | | | |
| $\alpha_1$ | Half-saturation uptake diatoms | 0.55 | mmol-N m$^{-3}$ |
| $\alpha_2$ | Half-saturation uptake flagellates | 0.45 | mmol-N m$^{-3}$ |
| $RD_0$ | Maximum uptake diatoms at 0ºC | 1.50 | d$^{-1}$ |
| $RF_0$ | Maximum uptake flagellates at 0ºC | 0.75 | d$^{-1}$ |
| $S_{DIA}$ | Sinking rate diatoms | -1 | m d$^{-1}$ |
| $Iopt_{dia}$ | Optimum PAR diatoms | 95 | W m$^{-2}$ |
| $Iopt_{flag}$ | Optimum PAR flagellates | 105 | W m$^{-2}$ |
| $k_c$ | Attenuation constant self-shading | 0.03 | m$^2$ (mg Chl a)$^{-1}$ |
| $LPN$ | Loss rate phytoplankton to nutrients at 0ºC | 0.03 | d$^{-1}$ |
| $LPD$ | Loss rate phytoplankton to detritus at 0ºC | 0.02 | d$^{-1}$ |
| $Ths_1$ | Half-saturation temperature diatoms | 12 | ºC |
| $Ths_2$ | Half-saturation temperature flagellates | 7 | ºC |
| $Q_{10}$ | Maintenance temperature coefficient | 0.07 | ºC$^{-1}$ |
| $RFR$ | Redfield ratio N:P (mol-based) | 16:1 | fraction |
| N:Si | Si:N-ratio (mol-based) | 1.1 | fraction |
| **Zooplankton** | | | |
| $Imax_{MEZ}$ | Maximum grazing mesozooplankton at 12ºC | 0.47 | d$^{-1}$ |
| $Imax_{MIZ}$ | Maximum grazing microzooplankton at 0ºC | 0.60 | d$^{-1}$ |
| $K_{MEZ}$ | Half-saturation ingestion mesozooplankton | 0.32 | mmol-N m$^{-3}$ |
| $K_{MIZ}$ | Half-saturation ingestion microzooplankton | 0.60 | mmol-N m$^{-3}$ |
| $AE_{MEZ}$ | Assimilation efficiency mesozooplankton | 0.65 | fraction |
| $AE_{MEZ}$ | Assimilation efficiency microzooplankton | 0.60 | fraction |
| $R_{MEZ}$ | Active respiration mesozooplankton | 0.29 | fraction |
| $R_{MIZ}$ | Active respiration microzooplankton | 0.35 | fraction |
| $\beta_{MEZ}$ | Basal respiration mesozooplankton at 0ºC | 0.005 | d$^{-1}$ |
| $\beta_{MIZ}$ | Basal respiration microzooplankton at 0ºC | 0.03 | d$^{-1}$ |
| $pref_{DI}$ | Grazing preference for diatoms by MEZ and MIZ | 1.0 | fraction |
| $pref_{FL}$ | Grazing preference for flagellates by MEZ and MIZ | 1.0 | fraction |
| $pref_{MIZ}$ | Grazing preference for microzooplankton by MEZ | 1.0 | fraction |
| $Mmax_{MEZ}$ | Maximum mortality mesozooplankton at 0ºC | 0.004 | d$^{-1}$ |
| $Mmax_{MIZ}$ | Maximum mortality microzooplankton at 0ºC | 0.030 | d$^{-1}$ |
| $KM_{MEZ}$ | Half-saturation mortality mesozooplankton | 0.07 | mmol-N m$^{-3}$ |
| $KM_{MIZ}$ | Half-saturation mortality microzooplankton | 0.02 | mmol-N m$^{-3}$ |
| $Ths_{MIZ}$ | Half-saturation temperature microzooplankton | 4 | ºC |
| SVM$_{MEZ}$ | Seasonal vertical migration mesozooplankton | 0-25 | m d$^{-1}$ |
| **Detritus and nutrients** | | | |
| $DN$ | Mineralisation of detritus at 0ºC | 0.001 | d$^{-1}$ |
| $DN_{Si}$ | Mineralisation of Si-detritus at 0ºC | 0.0001 | d$^{-1}$ |



| | | | |
|---|---|---|---|
| $NI_0$ | Maximum nitrification rate at 0 ºC | 0.02 | d$^{-1}$ |
| $K_{nit}$ | Oxygen half-saturation in nitrification | 3.75 | mmol-O$_2$ m$^{-3}$ |
| $K_{denit}$ | Nitrate half-saturation in denitrification | 0.135 | mmol-NO$_3$ m$^{-3}$ |
| $T_{sen}$ | Temperature coefficient on recycling processes | 0.07 | ºC$^{-1}$ |
| $SEDR$ | Sinking rate detritus | -20 | m d$^{-1}$ |
| RQN | Respiratory quotient in nitrification | 2.0 | O$_2$:NO$_3$ |
| RQC | Respiratory quotient in detritus | 1.0 | O$_2$:Organic-N |
| $S_{DET}$ | Settling rate detritus | 20 | m d$^{-1}$ |





## 10.2 Appendix B, the ocean model (HYCOM)

The ocean model (HYCOM) has 40 hybrid vertical levels, combining isopycnals with z-level
coordinates and sigma coordinates. Tides are included internally within the ocean model using
eight constituents and similar tides are added at the open boundaries using the Oregon State
University TOPEX/Poseidon Global Inverse Solution (TPXO 8.2,) Egbert and Erofeeva, 2002).
More than 100 rivers are included as monthly climatological discharges obtained from the
Global Runoff Data Centre (GRDC, http://grdc.bafg.de) and scaled as prescribed by Dai and
Trenberth (2002)(Dai and Trenberth, 2002). In addition the globally gridded Core v2 runoff data
(Large and Yeager, 2009) is added for Greenland, the Canadian Archipelago, Svalbard, and
islands within the Arctic Ocean.
The sea-ice model (CICE) describes the dynamics and thermodynamics of the sea-ice as
described by Rasmussen et al, 2018 (Rasmussen et al., 2018). The dynamics is driven by drag
from wind and ocean, surface tilt of the ocean, Coriolis force, and the internal strength of sea ice
that will resist movement of the ice pack. The internal strength is based on the Elastic-Viscous-
Plastic (EVP) sea-ice rheology (Hunke, 2001), that originates from the Viscous-Plastic (VP)
described by Hibler (1979)(Hibler, 1979). CICE includes 5 thickness categories of sea ice within
each grid cell in order to describe the inhomogeneity. The thermodynamics prescribes a vertical
temperature profile with a resolution of four sea ice layers and one layer of snow for each sea-ice
category (Bitz and Lipscomb, 1999). Snow is very important for the thermodynamics of sea ice
as it insulates sea ice from the atmosphere and has a higher albedo than sea ice. The lower
boundary is governed by the upper ocean temperature, which is usually the ocean freezing
temperature and is linearly dependent on its salinity. The upper boundary is governed by the heat
and radiation transfer between the atmosphere and the combined snow/ice surface. The net heat
flux is calculated based on the 2m atmospheric temperature, humidity, incoming long and short-
wave radiation, and 10m wind and the state of the surface of the sea-ice model.
The HYCOM and CICE models used in this paper are coupled on each time step using the Earth
System modeling Framework (ESMF) coupler (Collins et al., 2004). The HYCOM-CICE set-up
at DMI used in this paper covers the Arctic Ocean and the Atlantic Ocean, north of about 20°S,
with a horizontal resolution of about 10 km (Madsen et al., 2016)..





The HYCOM-CICE model system assimilates re-analyzed sea-surface temperature
(https://podaac.jpl.nasa.gov/GHRSST, Høyer et al., 2012, 2014) and sea-ice concentration
provided by the EUMETSAT Ocean and Sea Ice Satellite Application Facility (OSI SAF,
www.osi-saf.org, Lavergne et al., 2019) on a daily basis. The model is initialized in summer
1997 using the Polar Science Center Hydrographic Climatology (PHC; Steele et al., 2001) in the
Arctic Ocean and World Ocean Atlas 2001 0.25° (Conkright et al., 2002) in the Atlantic, with a
100 km linear transition. The atmospheric forcing is obtained from the Era-Interrim reanalysis
(Dee et al., 2011) until 2017 and thereafter deterministic HRES ECMWF forcing
(www.ecmwf.int).




**10.3 Appendix C, Figures**

Figure C1: Surface Chl *a* concentration (mg chl a m$^{-3}$) in 2010 obtained from the model (A-C) and from remote sensing (D-F). A) and D) are annual averages, B) and E) are April-June averages, and C) and F) are July-September averages.

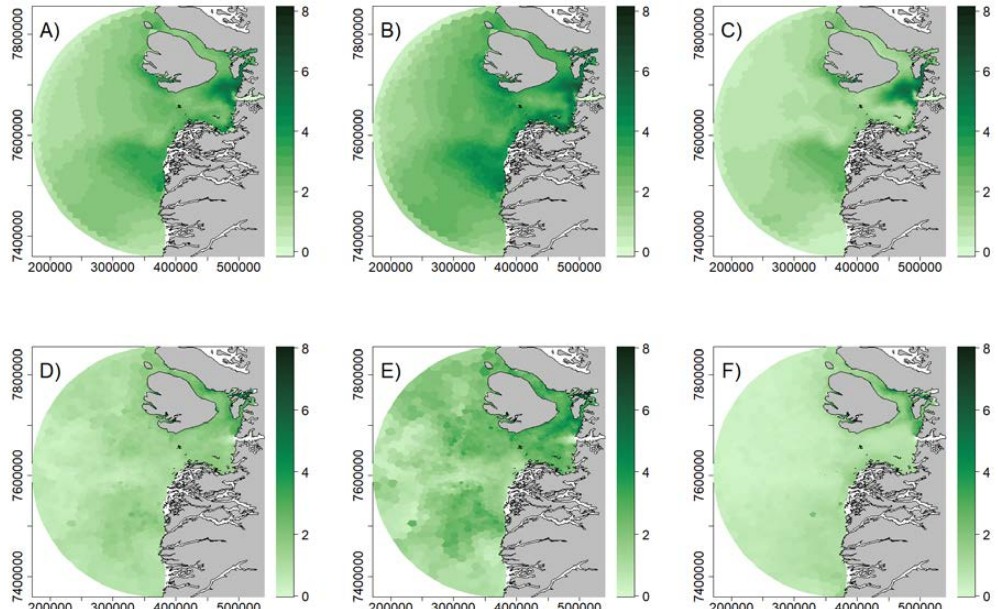







Figure C2: a) Position and b) bathymetry of transect (x-axis: distance in km, y-axis: depth in m) shown in Figure C3.

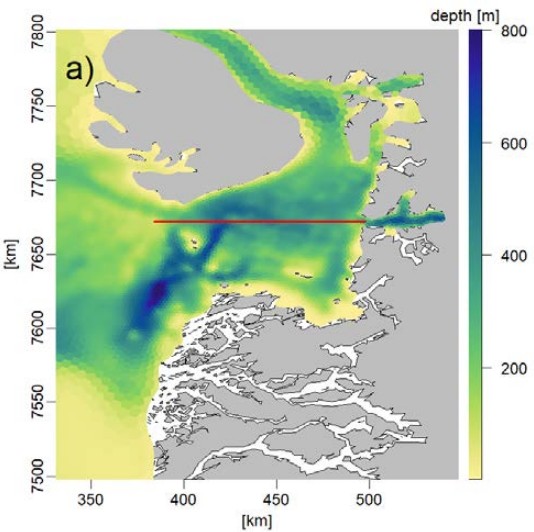

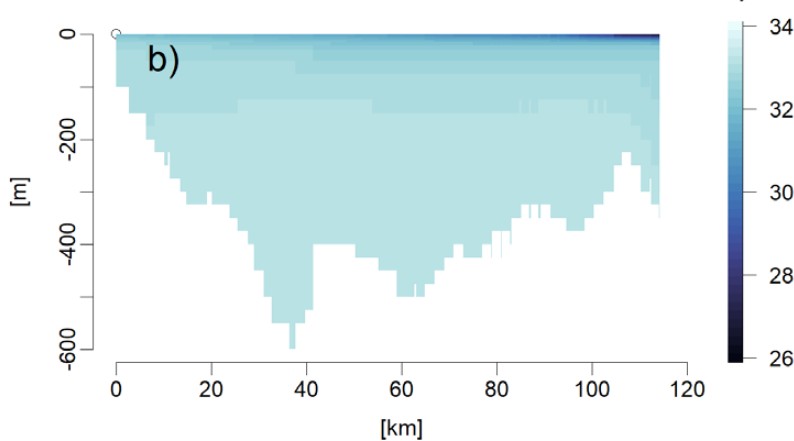






Figure C3: Transects (x-axis: distance in km, y-axis: depth in m) of salinity (a, b) temperature (°C) (c, d), DIN (mmol m$^{-3}$) (e, f), Chl $a$ (mg m$^{-3}$) (g, h) and NPP (mgC m$^{-3}$ d$^{-1}$) (i, j) in April (left) and August (right) 2010 along the transect shown in figure C2:

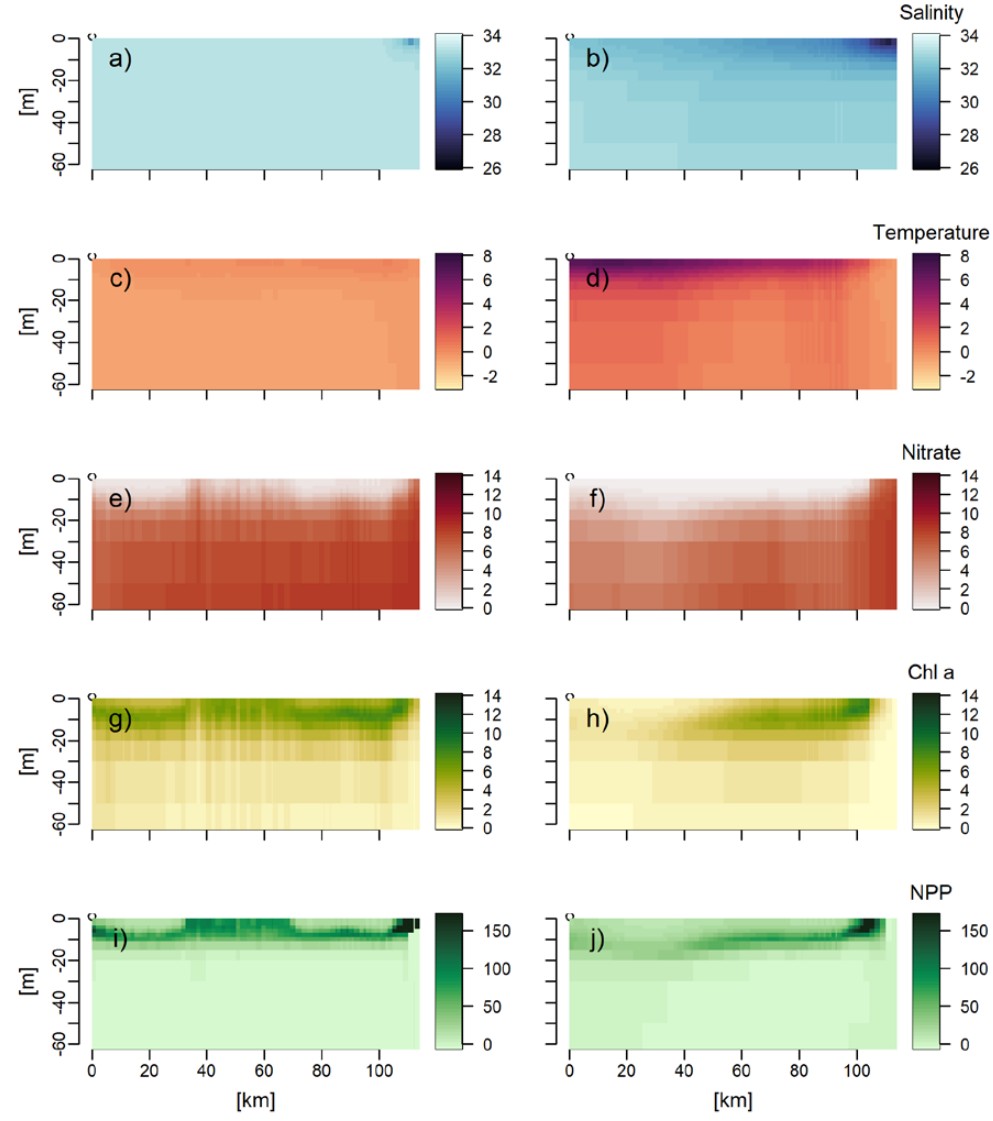
