# Peer review of "The sensitivity of primary productivity in"

_EGUsphere, 2022_

## Author Response (AR1)

**Reviewer 1**

The authors present with this paper a modelling approach to study the separate and cumulative effects of melt water runoff and sea ice cover on the net primary production in the Disko Bay area (West Geenland). This study highlights that sea ice reduction leads to changes in phytoplankton phenology as well as in an increase of the magnitude of primary production. It also showed that vertical mixing and tides have important role in nutrient replenishment in the area

I believe that the paper can be publish as it is. It is a simple and clear modelling experiment. The study design is clear and well justified. Authors accurately discussed every limitation of their approch in term of forcing, model parametrisation and spatial scale.

**Reply: Thank you**

 I only have few comments for the authors:

- L535 to L537, I think that the importance of the drivers is related to the spatial scale we are looking at. The region of the study  is quite large and the glacier discharges only influence a small area of the studied domain so it seems normal that the sea ice cover plays a stronger role in the control of the primary production.

**Reply: We agree that the spatial scale is of great importance, and this is also the message we try to convey, e.g. in the Abstract "Fresh water discharge had a strong local effect within ~25 km of the source sustaining productive hot spot during summer. When considering the annual NPP at bay scale, sea ice cover was the most important controlling factor." In the specific sentence mentioned (L535-37) we will like to emphasis also the temporal aspect, i.e. the effect of sea ice cover on spring timing production.**

- I noticed in the manuscript different spelling for 'sea ice', with and without hyphen. Strictly speaking, it should be spelled without hyphenation when it's a noun, and hyphenated when it is an adjective. In most in this manuscript sea ice is spelled without Hyphen and I will suggest to keep it like that everywhere else. L182, L194, L910, L930.

**Reply: we have checked the ms throughout, and corrected to "sea ice".**

**Reviewer 2**

The paper presents an interesting modelling exercise on the drivers and sensitivity of primary production to sea ice and freshwater runoff changes. Both in situ and remote sensing data are used to validation. The overall approach and the aim of the paper would be an important contribution to model primary production along the Greenland coast. However, there are two major issues that need to be addressed before the paper can be published. Firstly, I am very skeptical about the approach to not differentiate subglacial and surface runoff, since the effects for primary producers are very different. Secondly, most of the in situ data are not available and therefore the study is not reproducible. Some of the

cited papers do not supply their raw data, and the GEM database only has CTD data available, but none of the biogeochemical or sea ice data mentioned in the manuscript.

Subglacial upwelling: The authors do not differentiate between surface freshwater inputs and subglacial inputs. Their effects on coastal ecosystem and NPP are very different. Surface runoff introduces only few nutrients and leads to low surface salinity and strong stratification. Subglacial discharge introduces large amounts of NOx via subglacial upwelling, but the freshwater is highly diluted once it reaches the surface, if it reaches the surface at all. Consequently subglacial discharge leads to substantially weaker stratification. It is unclear if the model considers subglacial upwelling. The different sources should be considered separately, both in the discussion making clear what the effects of subglacial vs surface runoff are, in the model, and in the selection of reference stations. Currently there is no reference station or transect where surface runoff is the major freshwater source. Does this mean the manuscript focusses on the effects of subglacial discharge and upwelling? In this case this has to become clearer.

**Reply: This is the first attempt to model both the impact of the sea ice cover and the freshwater discharge in a coastal Arctic environment, using high resolution forcing factors. As stated in the ms there is room for further development by including higher resolution modelling particularly in front of the glacier. However, for the broad scale effects we are looking, this is not the first step. Subglacial discharge that enters at depth, will rise at the ice front within a short distance of 10s to 100s of meters from the ice front (Mankoff et al., 2016). Hence, this small-scale event occurs within the model grid cell at the run-off (1800 to 3200 m wide). In the model, we therefor inserted ice runoff in the surface layer. This approach creates a coastal upwelling and the effect of runoff that is described in the paper.**

**In the new version, we have included a test of the effect of instead inserting the ice runoff at the depth of the grounding line at the marine termination glaciers. These results are shown in the new Fig C4+C5. The effect is only minor (see below and in the revised ms for details).**

**Throughout, the revised manuscript we have focused on a clearer description of our current approach, and it limitations.**

Data availability: The authors refer to validation data from MarinBasis Disko of the GEM programme. However, only CTD data between 2011 and 2019 are available in the GEM database. No data for plankton communities or nutrients are reported. These data need to be publicly available to allow reproducibility of the study, either via updating the GEM database, or by achieving the data on another open access platform (e.g. PANGAEA). Other data are mentioned via citing earlier publications. However, many of these publications do not supply their raw data either. Consequently, the study is currently not reproducible. I highly recommend a data availability statement after making sure the raw data are publicly available. Ideally the model code should also be achieved (e.g. github, zenodo).

**Reply:  It is now clearly described were data can be assessed and additionally data and modelling code has been uploaded. References has been inserted in the manuscript.**

**CTD data from GEM can be found here: https://doi.org/10.17897/WH30-HT61**

**Sea Ice data from GEM can be found here:  https://doi.org/10.17897/SVR0-1574**

**The FlexSem source code and precompiled source code for Windows (GNU General Public License) can be downloaded at https://marweb.bios.au.dk/Flexsem. The specific code for the Disko set-up can be downloaded on Zenodo.org https://doi.org/10.5281/zenodo.7401870 (Larsen, 2022; Maar et al., 2022).**

**Data from Møller and Nielsen 2020 were uploaded to Zenodo.org https://doi.org/10.5281/zenodo.7454576 (Møller& Nielsen 2022)**

**The climatology from the current ms has been uploaded to Zenodo.org https://doi.org/10.5281/zenodo.7454727 (Møller et al 2022)**

Other comments:

L29: Fresh water should be written together as Freshwater

**Reply: This has been corrected throughout**

L30: A single productive hot spot needs an article "a" productive hot spot. OR productive hot spot"s"

**Reply: This has been corrected to "hot spot's".**

L60: wind-induced mixing needs a hyphen.

**Reply: This has been corrected**.

L67: deep subglacial upwelling would be mixed with large amounts of ambient seawater on the way up often not even reaching the surface. Thus, the effect if subglacially released freshwater on surface stratification is minor.

**Reply: the last part of the sentence has been deleted**

L78: their impacts in plural.

**Reply: This has been corrected**.

L85: a pronounced decrease needs an article.

**Reply: This has been corrected**.

L87: There do not seem to be any sea ice data in the GEM database. Please refer to the respective subprogram where the data are available.

**Reply: Since we originally downloaded the data there has been a restructuring of the database, and the data temporarily disappeared. The data is now available at https://doi.org/10.17897/SVR0-1574**

L133: there are issues with fixed elemental ratios in phytoplankton, especially under high light- low nutrient conditions (e.g. See Ross and Geider, 2009). This needs to be added to the discussion.

**Reply: We have added this to the discussion: "A more dynamic description of acclimation of primary productivity to different light and nutrient conditions (Ross and Geider, 2009), may be achieved by implementing variable element ratios (e.g., C:N) of phytoplankton instead of the fixed ratios in the current model."**

L148: the GEM database has does not show this relationship. Please refer to the specific report showing it or show the relationship in this manuscript in the supplement.

**Reply: We agree that this sentence was a bit unclear. We now specify that the general turbidity –salinity relationship <25 is from Murray et al, and the data from GEM are just used to set a constant for salinity >25. "Turbidity is strongly correlated with salinity and the background attenuation was described as a function of salinity: kdb=0.80-salinity x 0.0288 for salinity < 25 according to measurements across a salinity gradient in another Greenland fjord, the Young Sound (Murray et al., 2015) and set to a constant of 0.08 m-1 for salinity >25 according to monitoring data in the Disko Bay 69° 14' N, 53° 23' W (data.g-e-m.dk, https://doi.org/10.17897/WH30-HT61)."**

L167f: I see some issues with the assumption that all subglacial discharge will reach the surface layer in the fjord. Often the discharge reaches neutral buoyancy at depth. Also the resulting surface salinity and stratification is substantially lower compared to surface runoff due to the entrainment of saline bottom water. Overall, I am very skeptical to use the same model formulation for surface runoff (->low surface salinity, strong stratification, low nutrient inputs) and subglacial discharge (higher surface salinity, weak stratification, large amount of nutrient (mostly NOx) input by subglacial upwelling). See Hopwood et al. 2021. I recommend at least a model exercise separating these two different freshwater sources and check if the results differ substantially. One approach would be to add a reference station with high subglacial inputs and another station with high surface runoff. I would expect a larger positive effect on NPP with subglacial runoff (due to subglacial upwelling).

**Reply: In the model the resolution in the model does not allow us specifically model the fine scale plume dynamics in front of the glacier. Subglacial discharge that enters at depth, will rise up the ice front within a few 10s to 100s of meters of the ice front (Mankoff et al., 2016), this is within the grid cell at the ice boundary (1800 m wide). In the model we therefor**

inserted ice runoff in the surface layer. This approach creates a coastal upwelling, circulation and the effect of runoff that is described in the paper. In further development the model resolution can be increased if the focus is on the subglacial discharge, as also discussed in the paper.

Thus, in this version of the model it will not add any extra information to include an extra reference station, as suggested. We have instead included a scenario where we tested the effect of inserting the ice run off at the depth of the grounding line. This means that the subglacial discharge will rise a bit further from the glacier. These results are shown in Fig C4 and C5. The stratification and vertical distribution of nutrients, Chl and primary production are not changing much just establishing a bit further off in the late summer months (Fig C3+C5). The effect on the bay primary productivity is only minor (<1%).

We have, in the revised ms, tried to be clearer about our approach, the choice behind it and the potential implications.

L202: There are no sea ice data in the GEM database. Please refer to the respective subprogram where the data are available, or make sure they are publicly available elsewhere.

Reply: Since we originally downloaded the data there has been a restructuring of the database, and the temporarily disappeared.  The data is now available at https://doi.org/10.17897/SVR0-1574

L233f: Many of these validation data do not seem to be publicly available which makes the study not reproducible. The GEM database has only CTD data between 2011 and 2019. No data for nutrients, plankton biomass and communities, Chlorophyll are available in the GEM database. Please make sure the data are available and refer to the source. I recommend a data availability statement in the manuscript. Also the paper by Møller and Nielsen only shows the data in in plots and there seems to be no reference to the raw data.

Reply:

It is now clearly described were data can be assessed and additionally data and modelling code has been uploaded. References has been inserted in the manuscript.

CTD data from GEM can be found here: https://doi.org/10.17897/WH30-HT61

Sea Ice data from GEM can be found here:  https://doi.org/10.17897/SVR0-1574

The FlexSem source code and precompiled source code for Windows (GNU General Public License) can be downloaded at https://marweb.bios.au.dk/Flexsem. The specific code for the Disko set-up can be downloaded on Zenodo.org https://doi.org/10.5281/zenodo.7401870 (Larsen, 2022; Maar et al., 2022).

Data from Møller and Nielsen 2020 were uploaded to Xenodo.org
https://doi.org/10.5281/zenodo.7454576 (Møller& Nielsen 2022)

**The climatology from the current ms has been uploaded to Xenodo.org**
**https://doi.org/10.5281/zenodo.7454727 (Møller et al 2022)**

L287 and L289f: Here I see issues with the assumption that subglacial and surface runoff have the same effects on surface salinity (See comments above). Especially the station close to the Icefjord is very likely heavily influenced by subglacial upwelling of nutrients. I recommend adding a third reference station with high surface runoff and no subglacial inputs to check if the higher production is related to subglacial upwelling or the freshwater runoff.

**Reply: Please see the answer above**

L343: Would it be the same for surface runoff, or is the higher production with higher freshwater runoff from the glacier related to subglacial upwelling?

**Reply: As described above, and hopefully more clear from the revised ms, the current model describes the effect of the coastal upwelling, estuarine circulation and the effect of solid ice distributed in the upper 100 m, but not specifically the effect of subglacial upwelling. The effect of instead inserting the ice runoff at the depth of the grounding line caused a small spatial displacement of the primary production, and only had a small effect on the Bay primary productivity (<1%), as also described above.**

L400: Wind tides, but also subglacial upwelling are important nutrient sources in the described system. E.g. Juul-Pedersen et al. 2015 describes subglacial upwelling in Godthåbsfjord as the key driver for high late summer/autumn production in the entrance of the fjord system (ca 120 km distance from the glacier terminus).

**Reply: Juul-Pedersen et al suggest that the subglacial upwelling may play a role in the patterns found in the mouth of the fjord, but also discuss glacial runoff in general and the effect on fjord circulation. They conclude that "Further work is needed to separate how physical processes (i.e. glacier runoff and fjord circulation) and biological processes (i.e. species shifts and zooplankton grazing) influence the phytoplankton productivity and biomass." We have include a sentence referring to Juul-Pedersen et al "glacial runoff has been suggested to affect the seasonal development of phytoplankton 120 away from the glacier (Juul-Pedersen et al., 2015)."**

L433: Please add more details about the upwelling being enhanced by freshwater discharge. Do you refer to subglacial upwelling?

**Reply: Please see the answer above**

L434f: Higher nutrient concentrations in the surface water compared to the discharge is expected if subglacial upwelling is the source of nutrient inputs by bottom water entrainment. This shows that subglacial upwelling needs to be considered in the model.

**Reply: We see a clear effect of the freshwater runoff on the availability on nutrients and on the primary production (e.g. Fig 5) without specifically modelling the subglacial discharge, underlining the effect of the freshwater of the general circulation patterns e.g. coastal upwelling and estuarine circulation. As described in the ms further development of the model could test higher resolution at the plume.**

L464: I highly recommend excluding the 1996/1997 data. Without information about the method or access to the data it is not possible to reproduce the study or even judge the quality and limitations of the data. In situ bottle incubations can have a variety of issues and need a variety of metadata to calculate NPP in the fjord (e.g. light data, attenuation, Chlorophyll).

**Reply: We fully agree about the 1996/97 data, and the data from 1973-75 carries similar uncertainty. However, these are some of the only PP measurements available for the area, and we therefor still think it is worth referring to them.**

L512f: I am glad this issue is discussed, but I think that the main conclusion of this paper remain very speculative and poorly supported if subglacial and surface discharge are not treated separately. At the moment it is unclear if the positive effects of freshwater runoff are due to subglacial upwelling (subglacial discharge), or due to increased stratification (mainly surface discharge). I suggest checking if a reference station at a site with only surface runoff shows the same results as the current icefjord reference station. If yes, the conclusion of freshwater runoff as main driver (instead of subglacial upwelling), would be supported.

**Reply: We hope that our approach is clearer in the revised version of the manuscript. Please see the detail in the answers to the questions above.**

L515: The effects of subglacial upwelling can reach quite far (e.g. 120 km in Godthåbsfjord, where the tidewater glacier is smaller than in Illulisat, Juul-Pedersen et al., 2015).

**Reply: Yes, but Godthåbsfjord is a quite different system, it is a fjord system, whereas the Disko Bay are much more impacted by the open ocean. We have inserted a sentence referring to Juul-Pedersen et al 2015: "glacial runoff has been suggested to affect the seasonal development of phytoplankton 120 away from the glacier (Juul-Pedersen et al., 2015)." And furthermore a sentence underlining the difference between the two systems.**

L537: Freshwater is one word.

**Reply: This has been corrected.**

References:

Hopwood, M. J., Carroll, D., Dunse, T., Hodson, A., Holding, J. M., Iriarte, J. L., ... & Meire, L. (2020). How does glacier discharge affect marine biogeochemistry and primary production in the Arctic?. *The Cryosphere*, *14*(4), 1347-1383.

Juul-Pedersen, T., Arendt, K. E., Mortensen, J., Blicher, M. E., Søgaard, D. H., & Rysgaard, S. (2015). Seasonal and interannual phytoplankton production in a sub-Arctic tidewater outlet glacier fjord, SW Greenland. *Marine Ecology Progress Series*, *524*, 27-38.

Ross, O. N., & Geider, R. J. (2009). New cell-based model of photosynthesis and photo-acclimation: accumulation and mobilisation of energy reserves in phytoplankton. *Marine ecology progress series*, *383*, 53-71.

---

## Author Response (AR2)

The sensitivity of primary productivity in Disko Bay, a coastal Arctic ecosystem to changes in freshwater discharge and sea ice cover by Møller et al.

Thank you for the opportunity to further clarify. Please find below our answer to the comment by the reviewer.

**Reviewer**:
However, I am still surprised that subglacial discharge would have the same effects as surface discharge even on the large spatial scales of the model and the open system in Disko Bay. I agree that the discharge would still reach the surface (or near surface) layer within 10s to 100s of m, but the properties of this water would be very different from surface runoff. It would transport deep water nutrients with it which have been shown to affect nutrient dynamics and primary production on the mesoscale (100km, See Hopwood et al., 2020). The authors mention that surface runoff would create coastal upwelling, but I am not convinced that this would have the same effect as subglacial upwelling as described in numerous studies and fjord systems. If the authors disagree, I suggest adding a reference which shows that the effects of coastal and subglacial upwelling on nutrient concentrations can be comparable in a large open system such as Disko Bay. Is it possible that coastal upwelling is overestimated in the model in order to compensate for the lack of subglacial upwelling? I suggest discussing the source of nutrients in some more detail, or generalize the upwelling in your model as coastal upwelling, including subglacial upwelling. However, if coastal upwleling and subglacial upwelling are not comparable, I am not sure if the model can be generalized to systems with only surface runoff.

**Answer**: *We acknowledge the concerns of the reviewer and have therefore made a new analysis of the current velocities at the ice edge of the largest marine terminal glacier in the area, Jakobshavn Isbræ. We have plotted vertical profiles of the east-west velocity and the vertical velocity at the ice edge and found that the main outgoing transport from the ice fjord occurs in the bottom waters and that this leads to a coastal upwelling at the ice edge. Hence, even though the ice discharge is at the surface, it is fully mixed in the water column during transport from the release site at the mouth of the glacier to the ice edge. Due to ice cover, the wind driven transport is reduced in the surface layer and the main transport takes place near the bottom of the fjord. Hence, the model can to some extent reproduce the coastal upwelling from subglacial discharge at the edge of the marine terminating glacier and is able to produce realistic results despite the basin scale application and not fine scale modelling around the glacier. We have added a new figure C4 in the appendix showing the velocities and added new text to methods, results and discussion. We hope that the new text and figures clarify the questions raised.*

*Below the revised text in the different sections are inserted.*

*Section 2.6. Surface forcing data*

*Glacier ice cover was assumed to be present throughout the year in the Jakobshavn Isbræ near Ilulissat with the ice edge located at the mouth of the fjord whereas land- and ice runoff were located at the sub-arms of the fjord (Figure 1).*

*Section 3.5. Model scenarios with sea ice cover and discharge.*

*Horizontal (East-West) current velocity profiles at the ice edge (water depth of 241 m) of Jakobshavn Isbræ showed an outgoing westly direction with highest outflow at 150-200 m depth from March to October (Figure C4a). Vertical velocities showed an upward transport with highest values close to the bottom at 190-216 m depth (Figure C4b). The scenario with no runoff (noQNP) showed weaker horizontal transports and less upwelling at the ice edge (Figure C4). When ice run-off was released at*

*the glacier grounding line instead at the surface, only a small increase of horizontal and vertical velocities was found at 90-200 m depth relative to the baseline. In addition, a small spatial displacement of the primary production was seen (Fig C5). The stratification and vertical distribution of nutrients, Chl a and primary production were not changing much, just establishing a bit further offshore in the late summer months (Fig C3+C6). The effect on the bay primary productivity is only minor (<1%).*

***Chapter 4. Discussion.** We do not specifically model the subglacial discharge of freshwater from the marine terminating glaciers or from melting of the numerous large icebergs in the bay. Instead, the freshwater discharge from solid ice was distributed equally across the upper 100 m in the locations where marine terminating glaciers were present. Subglacial discharge that enters at depth, will rise up the ice front within a few 10s to 100s of meters of the ice front (Mankoff et al., 2016), which is within the grid cell size of the model. We therefor inserted ice discharge in the model surface layer that was found to be fully mixed in the water column during transport towards the ice edge. At the ice edge of the Jakobshavn Isbræ, modelled velocity profiles confirmed a bottom upwelling due to higher outgoing water transport at the bottom of the glacier (Figure C4a, b) in accordance with previous studies of marine terminating glaciers (Hopwood et al. 2020). In the scenario with no runoff (noQNP), the outgoing transport and vertical velocities at depths below 100m was severely reduced confirming the importance of ice discharge for the observed dynamic (Hopwood et al. 2020). When the discharge instead was inserted at the grounding line of the marine terminating glaciers, there was a limited increase in the vertical velocity marginal (Figure C5b). Similarly, there was only a slight displacement of the phytoplankton bloom to further offshore and very limited changes in the stratification and vertical distribution of nutrients, Chl a and NPP (Fig C5+C6). The effect of the primary productivity of the Bay was <1%.*

**New Figure C4.** Vertical profiles of a) East-West velocities and b) vertical velocities at the ice edge in Jakobshavn Isbræ for 2010, the scenario noQNP, and the scenario with subglacial discharge at the glacier grounding line

[Figure]